# Long-distance communication by specialized cellular projections during pigment pattern development and evolution

Dae Seok Eom[1], Emily J Bain[1], Larissa B Patterson[1], Megan E Grout[1], David M Parichy[1,2]*

[1]Department of Biology, University of Washington, Seattle, United States; [2]Institute for Stem Cell and Regenerative Medicine, University of Washington, Seattle, United States

**Abstract** Changes in gene activity are essential for evolutionary diversification. Yet, elucidating the cellular behaviors that underlie modifications to adult form remains a profound challenge. We use neural crest-derived adult pigmentation of zebrafish and pearl danio to uncover cellular bases for alternative pattern states. We show that stripes in zebrafish require a novel class of thin, fast cellular projection to promote Delta-Notch signaling over long distances from cells of the xanthophore lineage to melanophores. Projections depended on microfilaments and microtubules, exhibited meandering trajectories, and stabilized on target cells to which they delivered membraneous vesicles. By contrast, the uniformly patterned pearl danio lacked such projections, concomitant with Colony stimulating factor 1-dependent changes in xanthophore differentiation that likely curtail signaling available to melanophores. Our study reveals a novel mechanism of cellular communication, roles for differentiation state heterogeneity in pigment cell interactions, and an unanticipated morphogenetic behavior contributing to a striking difference in adult form.

*For correspondence: dparichy@uw.edu

**Competing interests:** The authors declare that no competing interests exist.

## Introduction

Genes contributing to phenotypic diversification are beginning to be identified, yet the morphogenetic mechanisms by which changes in gene activities are translated into species differences in form remain virtually unknown. Holding great promise for identifying such mechanisms are the diverse and ecologically important pigment patterns of fishes (*Endler, 1980*; *Houde, 1997*; *Wang et al., 2006*; *Engeszer et al., 2008*; *Price et al., 2008*; *Seehausen et al., 2008*; *Roberts et al., 2009*; *Kelley et al., 2013*) because pigment cell behaviors are observable as phenotypes unfold and because pigment pattern development is amenable to both experimental manipulation and theoretical modeling (*Painter et al., 1999*; *Kondo and Miura, 2010*; *Miyazawa et al., 2010*; *Caballero et al., 2012*; *Yamanaka and Kondo, 2014*; *Bullara and De Decker, 2015*; *Volkening and Sandstede, 2015*). In this regard, fishes of the genus *Danio* should be especially useful as their adult pigment patterns differ markedly among species and mechanisms of pattern formation are starting to be understood in zebrafish, *D. rerio* (*Parichy, 2015*; *Parichy and Spiewak, 2015*; *Singh and Nusslein-Volhard, 2015*; *Watanabe and Kondo, 2015*).

Zebrafish have dark stripes of black melanophores with light interstripes of yellow-orange xanthophores and iridescent iridophores (*Figure 1A*), all of which are derived from the neural crest, either directly, or through stem cell intermediates (*Budi et al., 2011*; *Dooley et al., 2013*; *Mahalwar et al., 2014*; *McMenamin et al., 2014*; *Singh et al., 2014*). Interactions among pigment

**eLife digest** Animals have very different patterns of skin pigmentation, and these patterns can be important for survival and reproduction. Zebrafish, for example, have horizontal dark and light stripes along their bodies, while a closely related fish called the pearl danio has an almost uniform pattern.

The dark stripes of the zebrafish contain cells called melanophores, while the lighter regions contain two other types of cells known as xanthophores and iridophores. These pigment cell types interact with each other to create stripes. The iridophores establish the lighter stripes and specify the position and orientation of the dark stripes. They also produce a protein called Csf1, which allows the xanthophores to mature. As the stripes form, melanophores present in lighter stripes move into nearby dark stripes. Pearl danios also contain these three types of pigment cells, but these cells remain intermingled giving the fish their uniform color.

Eom et al. have now used microscopy to image pigment cells in zebrafish and pearl danio to uncover how interactions between these cells differ in species with different pigment patterns. The technique involved tagging pigment cells with fluorescent markers and using time-lapse imaging to track them during the formation of the adult pigmentation pattern.

The experiments show that stripes form in zebrafish because the cells that make the xanthophores form long, thin projections that extend to neighboring melanophores. These so-called 'airinemes' deliver materials to melanophores and help to clear the melanophores from interstripe regions, partly by activating a cell communication pathway called Delta-Notch signaling. These cell projections are mostly absent from the cells that make xanthophores in the pearl danio due to differences in when Csf1 is produced. This alters the timing of when the xanthophores develop, leading to the loss of long-distance airineme signaling.

Eom et al.'s findings identify a new way in which cells can communicate and an unanticipated cell behavior that contributes to a striking difference in the pigmentation patterns of zebrafish and pearl danio. Future studies should further our understanding of these unique projections and reveal whether they are produced by other types of cells.

cells are essential to pattern formation (*Watanabe and Kondo, 2015*). Iridophores differentiate in the prospective interstripe, and specify the positions and orientations of melanophore stripes (*Frohnhofer et al., 2013*; *Patterson and Parichy, 2013*). Iridophores also promote the differentiation of xanthophores in the interstripe by expressing Colony stimulating factor-1 (Csf1) (*Patterson and Parichy, 2013*). Subsequently, interactions between cells of melanophore and xanthophore lineages are required during a period of stripe consolidation, in which some melanophores initially in the interstripe join the stripes, and, simultaneously, the stripe borders become increasingly organized (*Parichy et al., 2000*; *Maderspacher and Nusslein-Volhard, 2003*; *Parichy and Turner, 2003a*, *2003b*; *Quigley et al., 2005*; *Takahashi and Kondo, 2008*). Finally, this pattern is reiterated as additional iridophores invade the stripes, ultimately emerging on the other side, where they terminate the first stripes and initiate new interstripes and stripes dorsally and ventrally (*Patterson et al., 2014*; *Singh et al., 2014*).

Despite an increasing appreciation for the phenomenology of interactions among zebrafish pigment cells through cell transplantation, genetic analyses, laser ablation, in vitro manipulations, and theoretical modeling (*Maderspacher and Nusslein-Volhard, 2003*; *Nakamasu et al., 2009*; *Eom et al., 2012*; *Yamanaka and Kondo, 2014*), the cellular and molecular mechanisms relevant to these interactions in vivo remain largely unknown. Moreover, the ways in which such interactions may have changed during the development of naturally occurring, alternative pattern states across species have yet to be explored. Interesting in this context is pearl danio, *D. albolineatus*, which has a nearly uniform pattern of fewer melanophores, more xanthophores and an intermingling of all three pigment cell classes (*Figure 1A*) (*Quigley et al., 2005*; *Mills et al., 2007*; *Patterson et al., 2014*; *McCluskey and Postlethwait, 2015*).

Here, we show by time-lapse imaging of cells in their native tissue environment that stripe consolidation in zebrafish requires a novel class of fast, long cellular projection, extended by cells of the

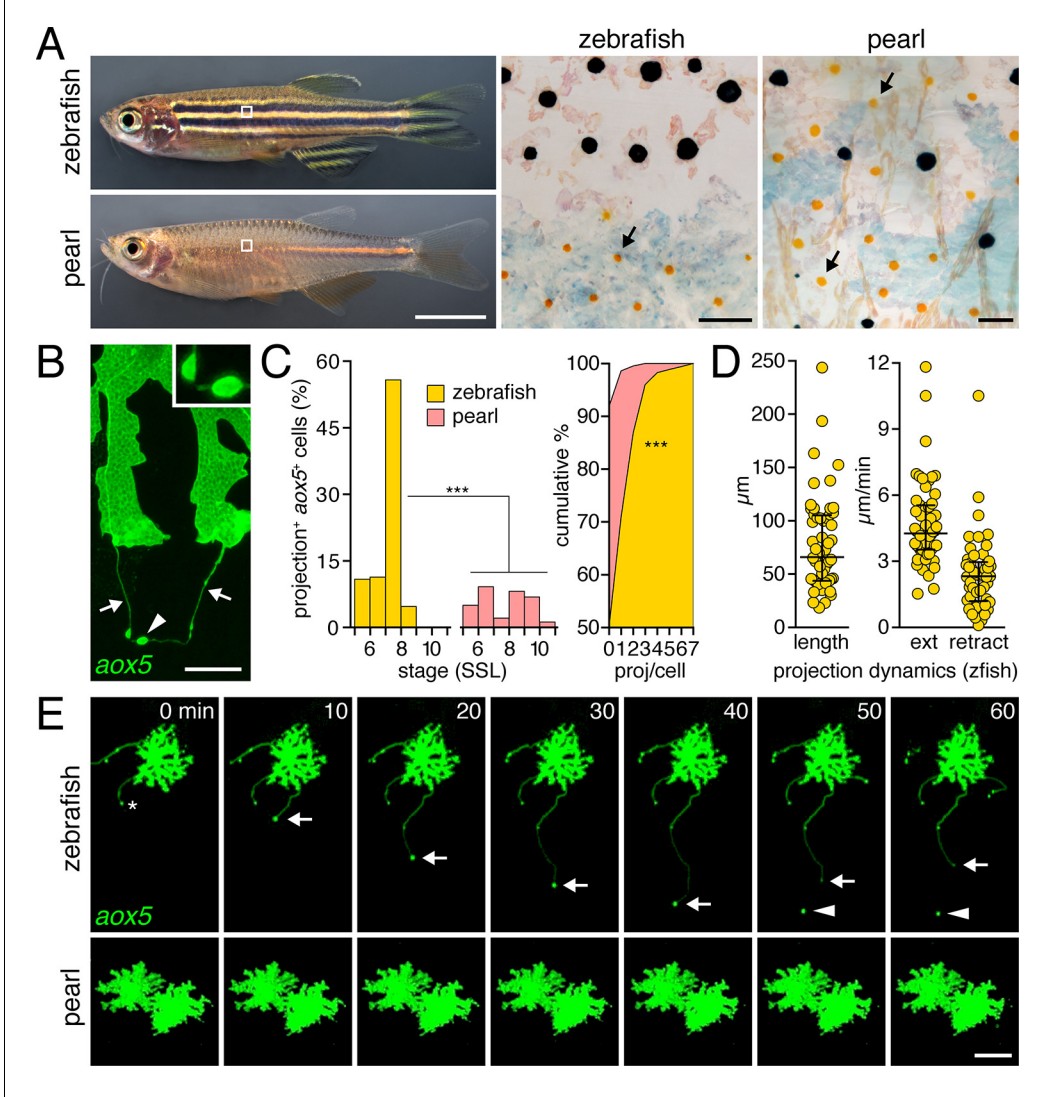

**Figure 1.** Pigment cell projections. (A) Zebrafish and pearl danio. Right, melanophores and xanthophores (arrows) after epinephrine treatment to contract pigment granules. (B) Long projections by zebrafish *aox5*+ cells of xanthophore lineage (arrows) with membraneous vesicles (arrowhead, inset). (C) Zebrafish *aox5*+ cells were more likely to extend projections than pearl, especially during early stripe development [7–8 SSL (*Parichy et al., 2009*); species x stage, $\chi^2$=103.4, d.f.=4, p<0.0001; N=929, 1259 cells for zebrafish and pearl; projections per cell: $\chi^2$=45.3, d.f.=1, p<0.0001]. (D) In zebrafish, projections were often long and fast. Bars indicate median ± interquartile range (IQR). (E) Extension and retraction (arrow) and release of vesicle (arrowhead) in zebrafish but not pearl. Scale bars: 5 mm (A, left); 50 μm (A, right); 10 μm (B); 50 μm (E).

The following figure supplements are available for figure 1:

**Figure supplement 1.** Developing adult pigment patterns during peak stages of *aox5*+ fast projections in zebrafish.

**Figure supplement 2.** Zebrafish *aox5*+ cells extend fast projections independently of melanophores and iridophores.

**Figure supplement 3.** Rare fast projections of zebrafish melanophores.

xanthophore lineage to melanophores. These projections, which we call 'airinemes,' contribute to transducing a Delta-Notch signal that promotes the clearance of melanophores from the developing interstripe. We further show that production and targeting of airinemes are differentiation-state dependent. Finally, using interspecific cell transplantation and transgenic manipulations we demonstrate that evolution of the very different, uniform pattern of pearl danio has entailed the loss of

long-distance airineme signaling, concomitant with modifications to xanthophore differentiation. Our results provide novel insights relevant to empirical and theoretical understanding of pigment pattern formation in zebrafish, as well as mechanisms of cellular communication and how they change during the evolution of adult phenotypes.

## Results

### Fast projections are extended frequently by cells of the xanthophore lineage in zebrafish but not pearl danio

To determine if species differences are associated with modifications to pigment cell interactions, we visualized cells using membrane-targeted fluorophores and time-lapse imaging during adult pigment pattern formation (*Budi et al., 2011*). In zebrafish, cells of the xanthophore lineage, marked by expression of *aldehyde oxidase 5* (*aox5*; formerly, *aox3*) (*Parichy et al., 2000*; *McMenamin et al., 2014*), exhibited long, fast projections with distinctive, membraneous vesicles at their tips; vesicles were often left behind when the projection carrying them retracted or fragmented (*Figure 1B,E*; *Video 1*). Such projections were especially frequent at early stages of stripe formation (7–8 SSL; *Figure 1C*-left; *Figure 1—figure supplement 1*) (*Parichy and Turner, 2003b*; *Parichy et al., 2009*; *Patterson and Parichy, 2013*) and individual cells could extend several projections over 18 hr of time-lapse imaging (*Figure 1C*-right). Projections reached as far as 5–6 cell diameters and extended and retracted quickly (*Figure 1D,E*). Incidences of projection formation were not markedly altered in mutants lacking melanophores or iridophores, or in transgenic fish having supernumerary melanophores (*Figure 1—figure supplement 2*; *Video 2*).

By contrast, fast projections were extended only rarely by *aox5+* cells of pearl danio (*Figure 1C, E*; *Figure 1—figure supplement 1 Video 1*). Likewise, fast projections were rare among melanophores of both species, marked by expression of *tyrosinase related protein 1b* (*tyrp1b*; *Figure 1—figure supplement 3*, *Video 3*) and were not observed for iridophores, marked by *purine nucleoside phosphorylase 4a* (*pnp4a*; *Video 4*) (*Lang et al., 2009*; *Curran et al., 2010*; *McMenamin et al., 2014*).

Thus, long, fast, and vesicle-containing projections were produced exuberantly by cells of the zebrafish xanthophore lineage during stages of stripe formation, but were not common to other pigment cell classes in this species or to cells of the xanthophore lineage in pearl danio.

### Fast pigment cell projections are distinct in cytoskeleton and morphology from long filopodia

We asked whether fast projections of pigment cells were similar to previously described cellular projections. Fast projections were considerably longer than typical (<10 μm) filopodia, that lacked vesicles (e.g., *Figure 1—figure supplement 3D*;

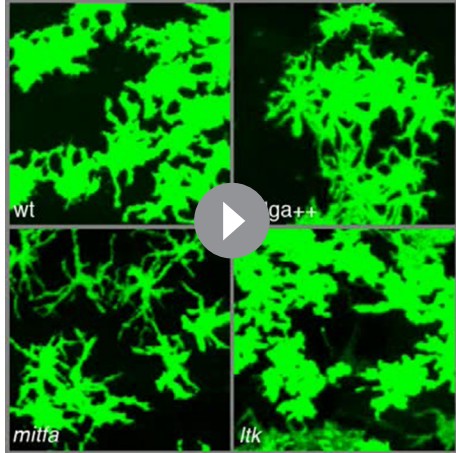

**Video 2.** Fast projections of *aox5+* cells persist in backgrounds with altered numbers of melanophores or iridophores. Shown are representative fields illustrating fast projections (arrows) of *aox5+* cells in the presence of excess melanophores (Kitlga++), an absence of melanophores (*mitfa*) and an absence of iridophores (*ltk*). 5 min interval, 375 min total.

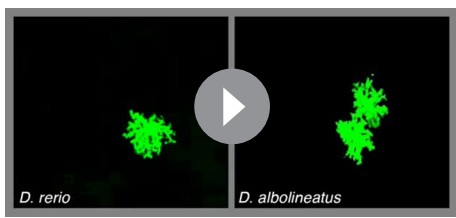

**Video 1.** Projections by cells of xanthophore lineage in zebrafish but not pearl. Left, in zebrafish *D. rerio*, an *aox5+* cell in a mosaically labeled larva extends fast, thin cellular projections. Right, such projections are not apparent in two *aox5+* cells in pearl. 10 min interval, 580 min total.

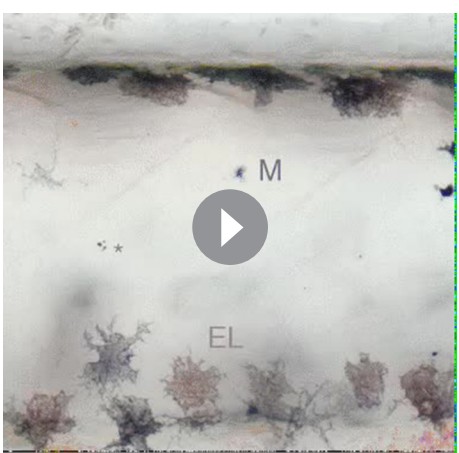

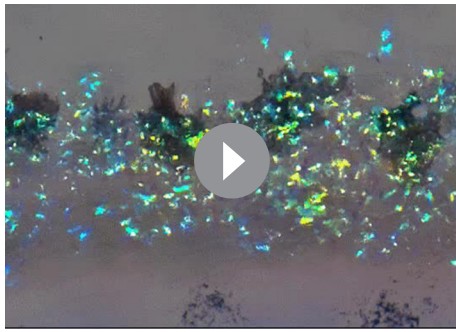

**Video 3.** Fast projections were only infrequently produced by cells of the melanophore lineage. Melanophores labeled with *tyrp1b*:membrane-mCherry. Most cellular processes were robust in size and relatively slow moving, although some could extend over long distances (yellow arrowhead). Only rarely were fast cellular projections extended (white arrow near top of frame). M, lightly melanized, differentiating adult melanophore in the prospective stripe region. EL, brownish embryonic/early larval melanophore persisting at the horizontal myoseptum in the prospective interstripe region. *, macrophage carrying mCherry+ debris. 5 min interval, 785 min total.

**Video 4.** Iridophores did not extend fast projections. Iridophores that have started to differentiate in the prospective interstripe (and in the vicinity of persisting EL melanophores), labeled with membrane-GFP driven by the promoter of *purine nucleoside phosphorylase 4a* (*pnp4a*). Thin, short and relatively straight projections are often extended at cell edges (e.g., arrow). An individual *pnp4a*+ cell is observed dispersing from the mat of aggregated *pnp4*+ iridophores (arrowhead). 5 min interval, 475 min total.

and below). Longer, often relatively straight, actin-based filopodia, or 'cytonemes,' function in intercellular communication in other systems (*Miller et al., 1995*; *Ramirez-Weber and Kornberg, 1999*; *De Joussineau et al., 2003*; *Cohen et al., 2010*; *Caneparo et al., 2011*; *Danilchik et al., 2013*; *Massarwa and Niswander, 2013*; *Sanders et al., 2013*; *Gradilla et al., 2014*; *Luz et al., 2014*; *Roy et al., 2014*; *Stanganello et al., 2015*). Similar to cytonemes, fast pigment cell projections contained F-actin, as evidenced by labeling with fluorescent reporters fused to the calponin homology domain of utrophin (UtrCH) as well as LifeAct (*Burkel et al., 2007*; *Riedl et al., 2008*; *Sanders et al., 2013*) (*Figure 2A,B*).

In contrast to cytonemes, however, fast projections had highly meandering trajectories and the vesicles associated with them had diameters of 1.7 ± 0.1 μm (mean ± SE, *n*=22), considerably larger than the 30–200 nm exosome-like particles trafficked within cytonemes (*Bischoff et al., 2013*; *Sanders et al., 2013*; *Gradilla et al., 2014*). Also unlike cytonemes, filaments of fast projections contained tubulin, as indicated by localization of an alpha tubulin fusion protein; such labeling was present in some but not all projection-associated vesicles (*Figure 2C*). Suggesting active microtubule assembly, fast projections and vesicles also exhibited transient accumulations of microtubule plus-end binding protein EB3 (*Figure 2D*; *Video 5*).

Fast projections were also 1–2 orders of magnitude longer than recently observed actin- and tubulin-containing projections of chick somitic epithelial cells (*Sagar et al., 2015*) and *Drosophila* germ line stem cells (*Inaba et al., 2015*), as well as cellular bridges that can transfer cytoplasmic materials between embryonic neural crest cells (*McKinney et al., 2011*). Finally, these projections differed markedly from long but slow and robust, pseudopodial-like processes sometimes extended by melanophores transiently during development (*Video 3*) and present in adult fish (*Hamada et al., 2014*).

Given their distinctive morphology and cytoskeletal composition, as well as their function (below), we termed these fast projections 'airinemes,' for Iris, who—fleet as the wind on golden wings—

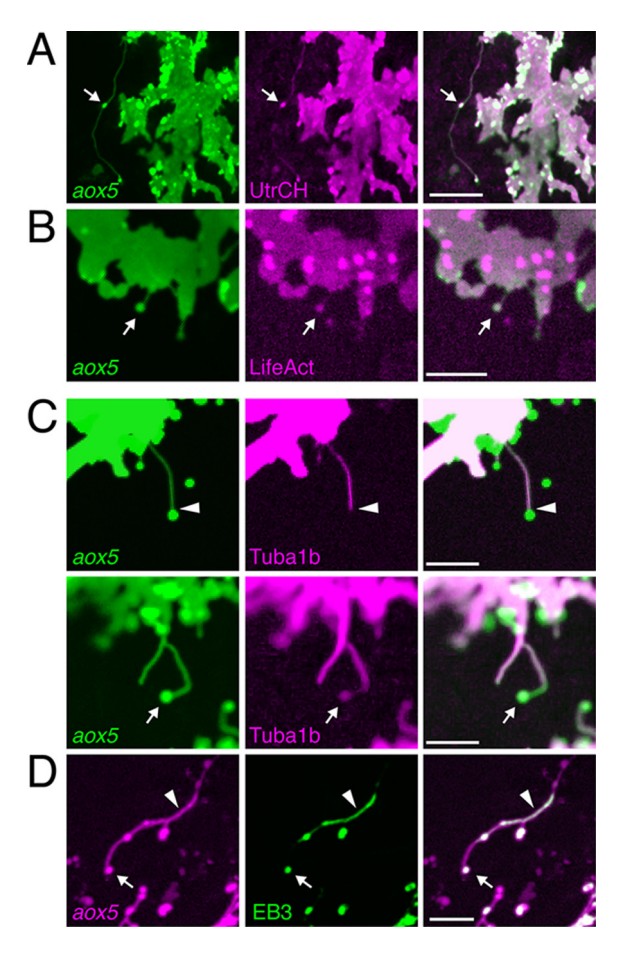

**Figure 2.** Fast projections harbor microfilaments and microtubules. (**A, B**) *aox5*+ projections and puncta (arrowheads) labeled for F-actin, revealed by UtrCH-mCherry (**A**) and LifeAct-mKate (**B**). (**C**) Projections contained tubulin, as revealed by Tuba1b-mCherry. In some instances tubulin was absent from vesicles (arrowhead, upper) and in other instances was found in vesicles (arrow, lower). (**D**) Accumulations of microtubule end binding protein EB3 fused to GFP (arrowhead), were present in vesicles as well (arrow); *aox5* here drives mCherry. Scale bars: 20 µm (**A**); 10 µm (**B, C, D**).

delivered messages for the gods (*Homer, ~8th cent. BCE*), as well as for Sir George Biddell Airy, who described limits on optical resolution (*Cox, 2012*).

## Airinemes are produced by xanthoblasts within melanophore stripes, rather than xanthophores within interstripes

As a first step towards understanding potential roles for airinemes in stripe formation, we sought to further characterize the cells that extend them. In zebrafish, *aox5*+ cells occur in the prospective interstripe, where they differentiate as xanthophores [beginning ~6.5 SSL (*Parichy et al., 2009*; *Patterson and Parichy, 2013*)]. Yet, *aox5*+ cells also occur at lower densities in developing and completed stripes, where they remain unpigmented or lightly pigmented (*Figure 3A*); we refer to these incompletely differentiated cells as xanthoblasts (*McMenamin et al., 2014*).

To determine if interstripe and stripe populations of *aox5*+ cells differ in airineme production we compared behaviors of cells in these locations during the peak of airineme deployment (7.5 SSL; *Figure 1C*). We found that cells within stripe regions, presumptive xanthoblasts, were more likely to extend airinemes than were cells within the developing interstripe, presumptive xanthophores (*Figure 3B*). Inspection of *aox5*+ cells at high resolution further revealed that unpigmented

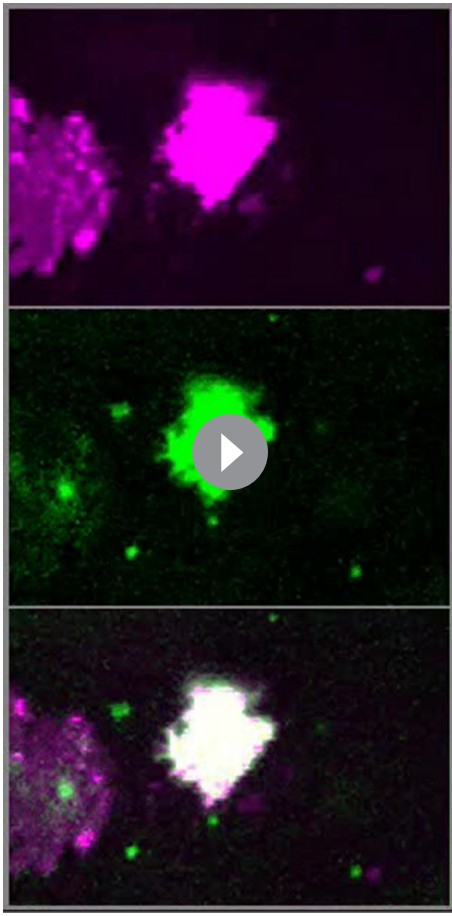

**Video 5.** Microtubule plus-end binding protein EB3 localized transiently in airinemes. Upper, membrane labeling with *aox5*:membrane-mCherry. Middle, EB3-GFP fusion protein driven by *aox5* promoter. Lower, merge. EB3-GFP is present transiently as airinemes extend (arrow). Green spots in background are accumulations of xanthophore pigment in neighboring cells. 5 min interval, 190 min total.

xanthoblasts, but not pigmented xanthophores, had membrane blebs from which airinemes originated (*Figure 3C*; *Videos 6*, *7*).

To test experimentally if airineme production was contingent upon differentiation state, we exploited the thyroid hormone (TH) dependence of xanthophore differentiation: *aox5*+ cells are differentiation-arrested in fish lacking TH, but differentiate fully across the flank in fish expressing excess TH (*McMenamin et al., 2014*). We found that *aox5*+ cells arrested as xanthoblasts (TH–) produced more airinemes than *aox5*+ cells forced to differentiate as xanthophores (TH++; *Figure 3D*; *Video 8*). These findings are consistent with the interpretation that airineme production is differentiation-state specific within the xanthophore lineage (and see below)

## Airinemes are target-specific and required for melanophore consolidation into stripes

To assess the significance of airinemes for pigment pattern formation we sought to block their production. Consistent with actin and tubulin cytoskeletal dependencies, airineme production was curtailed by acute treatment with the myosin II inhibitor blebbistatin (*Kovacs et al., 2004*) and the microtubule polymerization inhibitor nocodazole (*Figure 4—figure supplement 1A*), although longer term whole-fish treatments were lethal. By contrast, low levels of the Cdc42 small GTPase inhibitor ML141 (*Surviladze et al., 2010*) inhibited airineme production (*Video 9*) while allowing for extended whole-fish treatments. In larvae treated with ML141 through adult pigment pattern formation, melanophores occurred ectopically within the interstripe (*Figure 4—figure supplement 1A,B*).

To block airineme production specifically in the xanthophore lineage we constructed a TetGBD ('Tet') transgene (*Knopf et al., 2010*; *Patterson and Parichy, 2013*) driven by *aox5* to express dominant negative Cdc42$^{N17}$ (dnCdc42) (*Kieserman and Wallingford, 2009*), inducible in a temporally specific manner with dexamethasone and doxycycline (dd; *Figure 4—figure supplement 2A*). Cdc42 has roles in cytoskeletal organization and a variety of cell behaviors and its inhibition blocks filopodial extension in other systems (*Etienne-Manneville, 2004*; *Wu et al., 2008*; *Chen et al., 2012*; *Sadok and Marshall, 2014*; *Stankiewicz and Linseman, 2014*). We found that extended, low-level induction of dnCdc42 in *aox5*+ cells inhibited airineme production (*Figure 4A*; *Video 10*) without significantly affecting the production of short filopodia, lamellipodia-like protrusive activities, the numbers of differentiated xanthophores or melanophores, or the distributions of *aox5*+ cells (*Figure 4—figure supplement 2B–F*). Yet, in *aox5*:Tet:dnCdc42 fish inhibited for airineme production, melanophores persisted ectopically in the interstripe (*Figure 4B*), as for whole fish treated with ML141.

These findings suggested that airinemes promote the consolidation of melanophores into stripes. During normal stripe development embryonic melanophores persisting from the early larval pigment pattern occur near the horizontal myoseptum, within the prospective interstripe. Many of these cells move short distances to join the developing stripes (*Parichy et al., 2000*; *Parichy and Turner,*

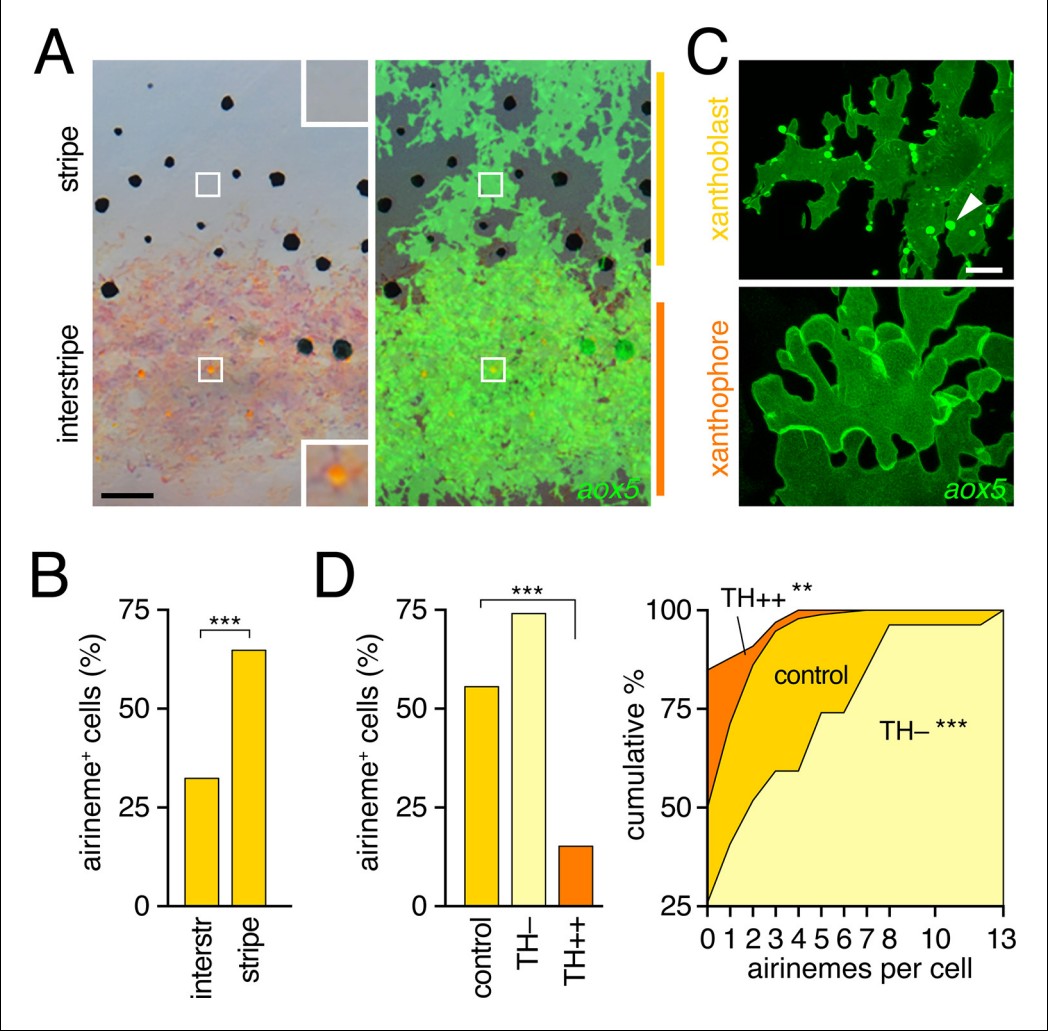

**Figure 3.** Airinemes were produced by xanthoblasts within prospective stripe regions. (**A**) Detail of developing pattern in zebrafish, illustrating pigmented *aox5*+ xanthophores of the interstripe as well as unpigmented *aox5*+ xanthoblasts of the prospective stripe (insets, boxed regions shown at higher magnification). Larva shown is 8.6 SSL, when xanthophore pigment is more readily visible, but after the peak of airineme production (*Figure 1C*). (**B**) *aox5*+ cells in prospective stripe regions were more likely to extend airinemes than *aox5*+ cells of the interstripe ($\chi^2$=28.6, d.f.=1, p<0.0001, N=295 cells). (**C**) Xanthoblasts (upper) had numerous membrane blebs (arrowhead), whereas xanthophores (lower) had smooth surfaces and more lobular edges (7.8 SSL). (**D**) Forced differentiation (TH++) reduced the incidence of cells that extended airinemes (left; $\chi^2$=12.2, d.f.=1, p<0.0001, N=123 cells) and the numbers of airinemes extended by each cell (right; TH++; $\chi^2$=12.0, d.f.=1, p<0.05), whereas differentiation-arrest increased airineme production (TH–; $\chi^2$=29.6, d.f.=1, p<0.0001; 7.5 SSL) Scale bars: 50 μm (**A**); 10 μm (**C**).

2003b; *Takahashi and Kondo, 2008*; *Patterson and Parichy, 2013*; *Patterson et al., 2014*), and their brownish color makes them distinguishable from gray–black adult melanophores that differentiate post-embryonically (*Quigley et al., 2004*; *Parichy and Spiewak, 2015*). In wild-type controls, these brownish embryonic melanophores had translocated to the edges of adult stripes, whereas in *aox5*:Tet:dnCdc42-expressing fish these melanophores remained in the interstripe (insets, *Figure 4B*).

Given these observations, we predicted that *aox5*+ airinemes interact directly with melanophores and so we time-lapse imaged *aox5*+ cells simultaneously with melanophores marked by *tyrp1b*. In these larvae we observed frequent, prolonged stabilization of *aox5*+ airinemes on melanophores (*Figure 4c,d*-left; *Video 11*). In contrast, airinemes retracted rapidly after contacting *aox5*+ cells (*Figure 4d*-left), indicating target specificity.

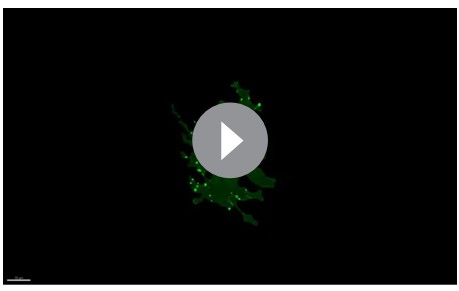

**Video 6.** Membrane blebs of xanthoblast. Shown is a static image of a single *aox5+* xanthoblast within the prospective stripe region, illustrating numerous membrane blebs, limited primarily to the superficial (epidermal-facing) surface.

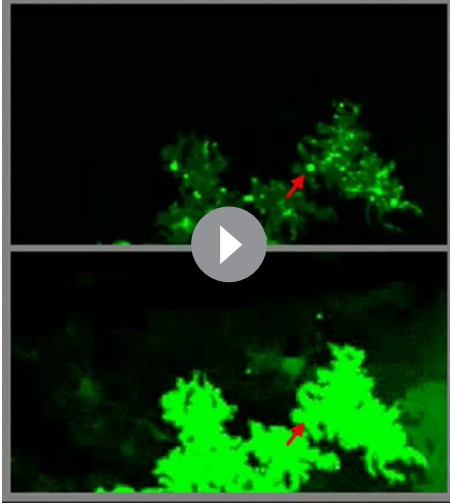

**Video 7.** Airinemes arose from membrane blebs of xanthoblasts. Shown are corresponding views of same field exposed to highlight membrane blebs from which airineme originates (upper, red arrow) and airineme filament (lower). An additional bleb formed near the end of observation (yellow arrow, upper). 5 min interval, 300 min total.

Airineme targeting was further specific to sub-populations of melanophores. In addition to persisting embryonic melanophores ($M_e$), newly differentiating adult melanophores ($M_n$) join the stripes, whereas previously differentiated adult melanophores ($M_p$) occur already within the stripes (*Figure 4C*) (*Parichy et al., 2000*; *Parichy and Turner, 2003b*; *Takahashi and Kondo, 2008*; *Patterson and Parichy, 2013*). Airinemes often stabilized on $M_e$ and newly differentiating, still migratory adult melanophores ($M_n$), but rarely on previously differentiated adult melanophores ($M_p$) (*Figure 4E*; *Figure 4—figure supplement 3*; *Videos 11, 12*). Moreover, airineme-delivered vesicles often remained on melanophores long after the airineme itself (*Figure 4G*; *Video 13*). In some instances, vesicles were transferred without evident airinemes (*Video 14*), a behavior that was also abrogated by dnCdc42.

To test whether *aox5+* airineme targeting is genetically labile, we screened zebrafish pigment pattern mutants and also re-examined the very few airinemes produced in pearl danio. As compared to wild-type zebrafish, we found that airinemes in spotted *connexin41.8 (cx41.8; leopard)* mutants (*Maderspacher and Nusslein-Volhard, 2003*; *Watanabe et al., 2006*) were later-appearing and more promiscuous, stabilizing as extensively on *aox5+* cells as on melanophores (*Figure 4D*-right, 4F; *Figure 4—figure supplement 4A,B*; *Video 15*). Such differences could reflect acute morphoge-netic interactions, perhaps dependent on connexin hemichannels, or chronic defects in differentiation, as xanthophores of presumptive *cx41.8* mutants are reported to contain melanin and melanosome-like vesicles (*Kirschbaum, 1975*). Interestingly, the rare airinemes of pearl danio also stabilized on xanthophores though inspection *cx41.8* sequence and expression did not reveal gross differences between zebrafish and pearl (*Figure 4D*-right; *Video 16*; and data not shown).

Together these results suggest that, in wild-type zebrafish, airinemes extended by *aox5+* xanthoblasts in prospective stripe regions contact melanophores still in the interstripe and promote their clearance during stripe consolidation, and that specificity of airineme targeting is genetically (and evolutionarily) labile.

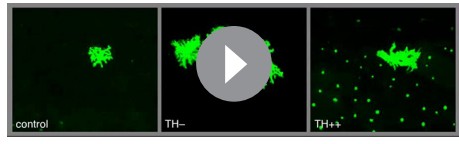

**Video 8.** Airineme production was differentiation-state dependent. Left, a wild-type control *aox5+* cell extended several airinemes. Middle, exuberant airineme production by differentiation-arrested *aox5+* cells in hypothyroid (TH–) fish in which the thyroid had been ablated transgenically at 4 days post-fertilization (McMenamin et al., 2014). Right, forced *aox5+* cell differentiation in hyperthyroid (TH++) mutant *(opallus, tshr^{D632Y})* resulted in failure of airineme production. Spots of autofluorescence are evident in neighboring cells lacking *aox5:membrane-GFP*. 5 min interval, 795 min total.

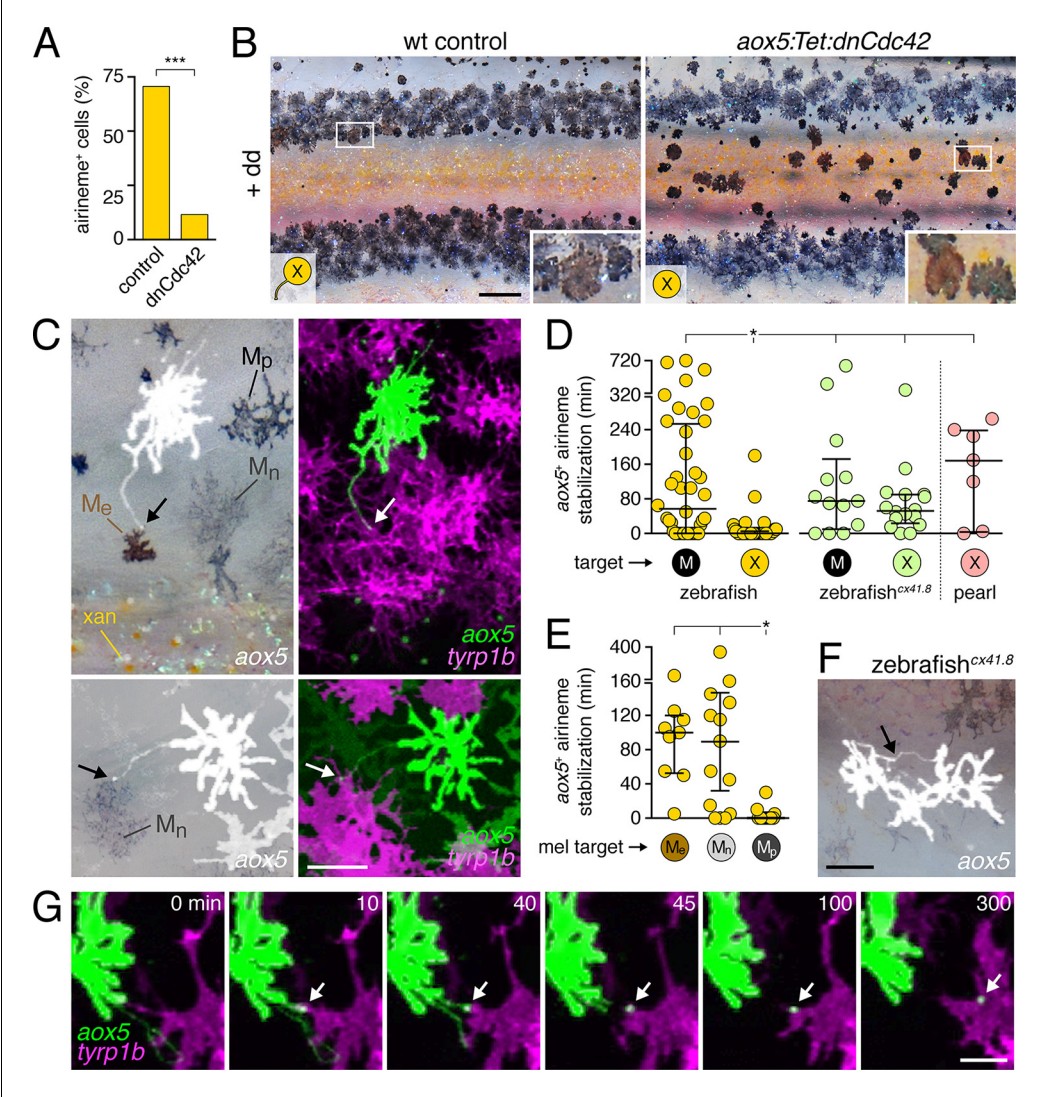

**Figure 4.** Airineme dependent patterning and airineme targeting specificities. (**A**) dnCdc42 blocks airineme extension ($\chi^2$=16.4, d.f.=1, p<0.0001, N=43 cells total). (**B**) Interstripe melanophores persisted when airinemes were blocked with dnCdc42. Cell states are indicated by logos in lower left corners (X, *aox5*+ xanthophore lineage). Insets, brownish melanophores persisting from embryonic/early larval pattern and gray–black adult melanophores. (**C**) Airinemes contacting melanophores (arrows; $M_e$, early larval; $M_n$, new; $M_p$, previously differentiated; xan, xanthophore). (**D**) Stabilization times (median ± IQR) of *aox5*+ airinemes on cells of melanophore (M) or xanthophore (X) lineages for zebrafish, *cx41.8* mutant zebrafish, and pearl danio. *aox5*+ airinemes of wild-type zebrafish were less likely to stabilize, and stabilized more briefly (*, both p<0.0001) after contacting cells of the xanthophore lineage as compared to melanophores; this target specificity was altered in *cx41.8* mutant zebrafish as well as pearl danio. Y-axis is split for clarity. (**E**) Zebrafish *aox5*+ airinemes were most likely to stabilize on $M_e$ and $M_n$ (*, p<0.0001; median ± IQR). (**F**) *cx41.8* mutant airinemes stabilized on *aox5*+ cells (arrow). (**G**) Vesicle transfer (arrow) to melanophore. Scale bars: 200 µm (**B**); 50 µm (**C**); 50 µm (**F**); 25 µm (**G**).

The following figure supplements are available for figure 4:

**Figure supplement 1.** Pharmacological blockade of airineme production.

**Figure supplement 2.** Expression of dnCdc42 transgene for inhibition of airinemes.

**Figure supplement 3.** Prolonged airineme contact with motile melanophores.

**Figure supplement 4.** Connexin dependence of airineme production.

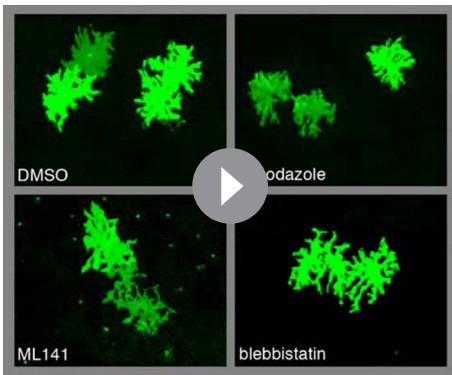

**Video 9.** Pharmacological inhibition of airineme production. Representative fields in which fast projections are extended by *aox5+* cells in DMSO-treated controls, but not during acute administration of nocodazole, ML141 or blebbistatin. Note that cells continue to change shape and extend some slow moving processes, similar to controls. Stationary fluorescent puncta are pigment autofluorescence in neighboring cells lacking *aox5*:membrane-GFP. 5 min interval, 450 min total.

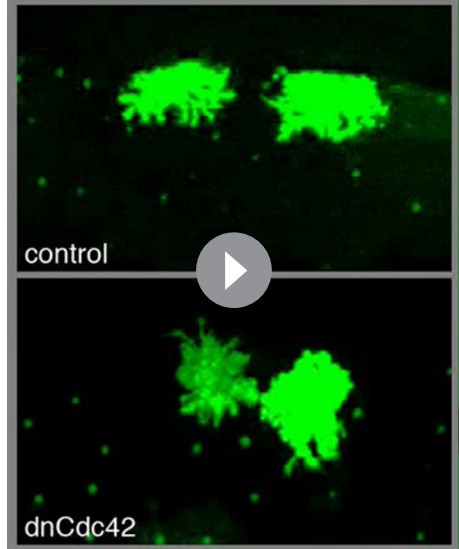

**Video 10.** Airineme production was inhibited by dnCdc42. Upon treatment with dd, non-transgenic control cells extended airinemes normally, whereas cells in fish transgenic for *aox5:Tet:dnCdc42* failed to extend airinemes despite forming other processes. Stationary fluorescent puncta are pigment autofluorescence in GFP– cells. 5 min interval, 415 min total.

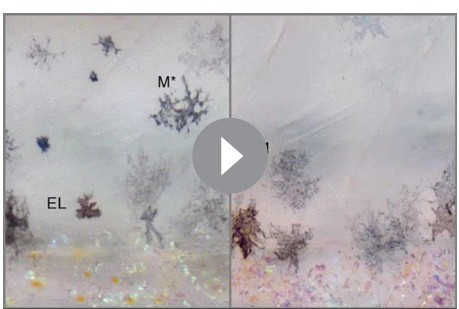

**Video 11.** Airinemes extended by cells of the xanthophore lineage stabilized on melanophores. Shown are prospective dorsal stripe regions of two individuals; prospective interstripe is at bottom of frame. In each view, an *aox5+* cell (green) extends an airineme (arrow) that stabilized (*) on a nearby *tyrp1b*:mCherry+ melanophore (magenta). EL, persisting embryonic/early larval melanophore. M, lightly pigmented adult melanophore. M*, more heavily pigmented adult melanophore. Video shows brightfield views before and after time-lapse. 5 min interval, 475 min total.

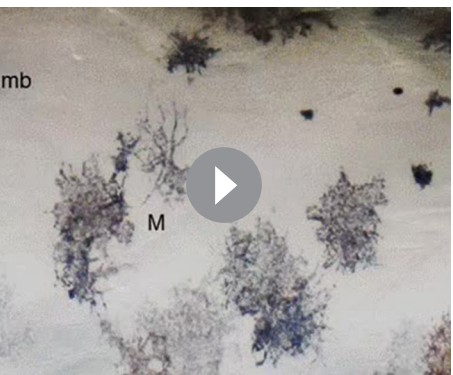

**Video 12.** Airineme targeting differed among melanophore subpopulations. Several previously differentiated *tyrp1b+* melanophores (M; magenta) are shown as well as an initially unpigmented *tyrp1b+* melanoblast (mb, upper left). Although numerous airinemes were extended by *aox5+* cells in the vicinity of melanophores, only one (yellow arrow) stabilized (*) on a previously differentiated melanophore. Another airineme stabilized on the differentiating melanoblast (yellow arrowhead), which received additional *aox5+* puncta. Video hows brightfield views before and after time-lapse and illustrates acquisition of melanin by as the melanoblast continued to differentiate. 5 min interval, 905 min total.

**Video 13.** Airineme-delivered membraneous vesicles persisted on melanophores. Left, *aox5+* cell (green) extending airineme. Middle, *tyrp1b+* melanophores (magenta). Right, merge. After the airineme contacted the melanophore it retracted, leaving behind a GFP+ vesicle that persisted on the melanophore membrane (arrow). 5 min interval, 310 min total.

**Video 14.** Membrane vesicles could be transferred in the apparent absence of airinemes. Left, *aox5+* cell (green). Middle, *tyrp1b+* melanophores (magenta). Right, merge. Several vesicles are indicated by yellow and white arrows. 5 min interval, 325 min total.

## Airinemes promote Delta–Notch signaling necessary for stripe consolidation

We sought to understand molecular bases for *aox5+* airineme effects on melanophore patterning in zebrafish. A good candidate for mediating such interactions is Delta-Notch signaling. In amniotes, Notch promotes melanocyte migration and survival and also regulates differentiation (*Moriyama et al., 2006*; *Schouwey et al., 2007*; *Aubin-Houzelstein et al., 2008*; *Kumano et al., 2008*; *Pinnix et al., 2009*). In zebrafish as well as pearl danio, xanthophores express Delta genes whereas melanophores express Notch genes (*Hamada et al., 2014*) (*Figure 5—figure*

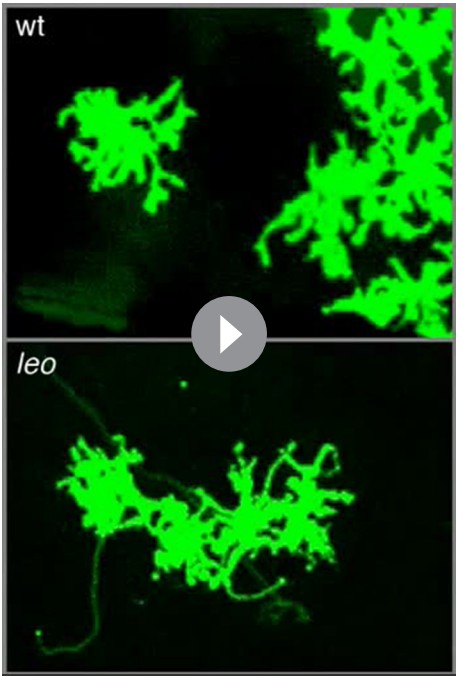

**Video 15.** Altered airineme targeting altered by connexin mutation. Upper, wild-type zebrafish airinemes (white arrow) retracted rapidly after contacting other *aox5+* cells. Lower, *leopard* (*cx41.8*) mutant airinemes (red arrow) frequently stabilized on other *aox5+* cells. 5 min interval, 450 min total.

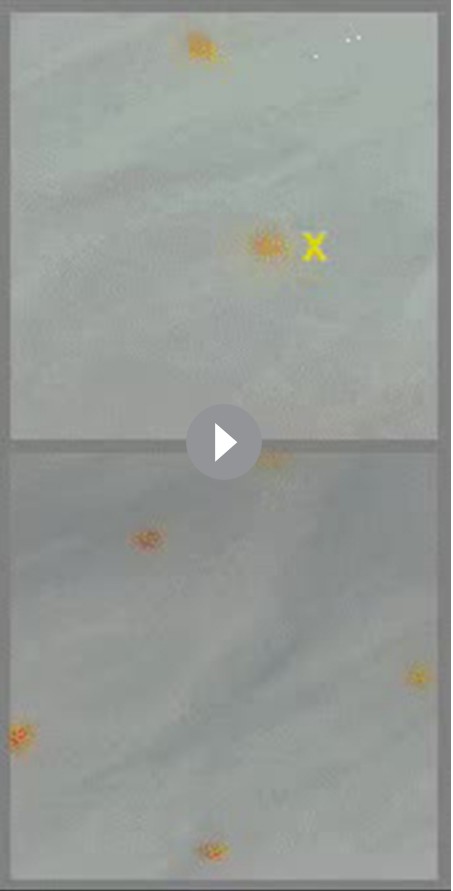

**Video 16.** Rare airinemes projected by pearl *aox5+* cells stabilized on xanthophores. Representative airinemes (arrowheads) stabilized on xanthophores, marked by orange pigment granule at cell centers (e. g., x). 5 min interval, 385 min total.

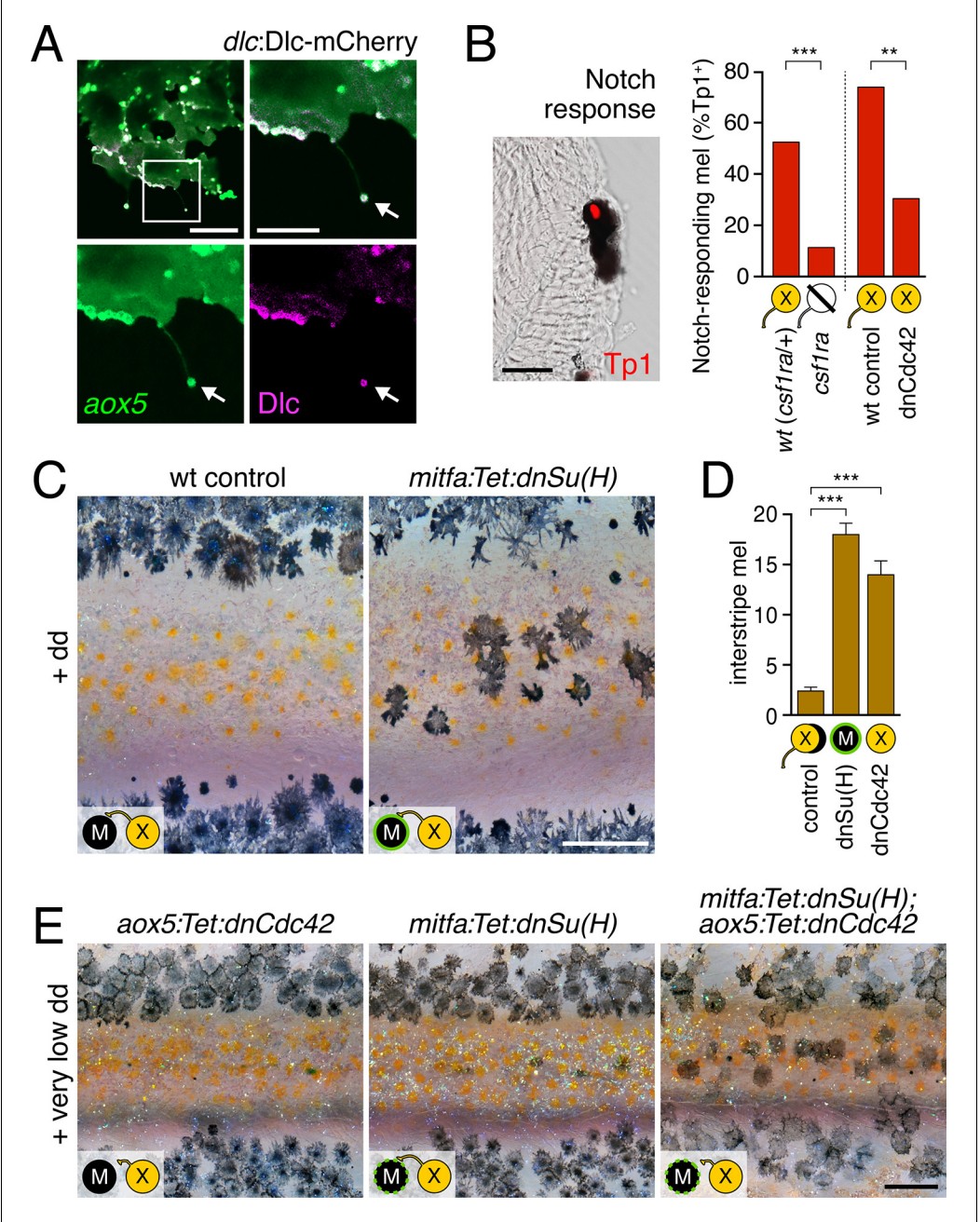

**Figure 5.** Melanophore clearance requires airineme-dependent Notch signaling in melanophores. (A) *aox5+* airineme vesicles harbored Dlc-mCherry (arrow). (B) Frequencies of Tp1+ Notch-responding melanophores were reduced in the absence of *aox5+* cells (*csf1ra*; $\chi^2$=15.5, d.f.=1, p<0.0001, N=77 cells) or airinemes (dnCdc42; $\chi^2$=9.0, d.f.=1, p<0.005, N=46 cells). Epidermal and muscle cells were also Tp1+ (not shown) but within the hypodermis, where pigment cells reside (*Hirata et al., 2003*), we observed only melanophores to be Tp1+. (C) Interstripe melanophores persisted when Notch signaling was blocked within the melanophore lineage by dnSu(H) [melanophore logo with green outlining], similar to dnCdc42 blockade of *aox5+* airinemes (compare with *Figure 4B*). (D) Quantification of interstripe melanophores (mean ± SE) in dd-treated non-transgenic fish as well as *mitfa*:Tet: dnSu(H) and *aox5*:Tet:dnCdc42 (overall, $F_{2,12}$=58.6, p<0.0001). (E) When induced with threshold levels of dd, persisting interstripe melanophores (means ± SE) were threefold more abundant ($t_7$=2.9, p<0.05) in fish doubly transgenic for *mitfa:Tet:dnSu(H)* and *aox5:Tet:dnCdc42* as compared to singly transgenic fish (shown at 9.8 SSL). Scale bars: 15 μm (A, left); 10 μm (A, right); 20 μm (B); 200 μm (C); 200 μm (E).

The following figure supplements are available for figure 5:

**Figure supplement 1.** Delta-Notch gene expression and patterning consequences.

*Figure 5 continued on next page*

*supplement 1A*). Consistent with a role for Notch in promoting melanophore migration, we found that expressing the constitutively active Notch1a intracellular domain in melanophores, using a promoter derived from the *microphthalmia a (mitfa)* gene (*Lister et al., 1999*), resulted in stripes that were significantly broader than in wild type (*Figure 5—figure supplement 1B*). Conversely, *delta c (dlc)* mutants exhibited melanophores ectopically in the interstripe, similar to the *aox5*:Tet:dnCdc42 phenotype (*Figure 5—figure supplement 1C*).

We predicted that if airinemes contribute to transducing a Delta-Notch signal from xanthoblasts to melanophores, then airinemes or their associated vesicles should harbor Delta proteins and perturbations to airineme production should impair Notch signaling within melanophores. We recombineered a 109 kb BAC containing zebrafish *dlc* coding sequence and regulatory elements to generate an mCherry fusion that was functional in rescuing *dlc* knockdown phenotypes; static and time-lapse imaging confirmed that DlC-mCherry was present in airineme vesicles (*Figure 5A*; *Video 17*). Technical limitations precluded corresponding analyses for a second Delta ligand expressed by xanthophores, Dll4 (*Figure 5—figure supplement 1A*).

To assess Notch responsiveness in melanophores, we used a synthetic Notch signaling activity detector, Tp1, consisting of multiple RBP-Jk-binding sites and a minimal promoter to drive fluorescent reporters (*Parsons et al., 2009*; *Ninov et al., 2012*). Destabilized and stabilized reporters were insufficiently sensitive to allow reliable real-time monitoring of signals upon airineme contact (not shown), yet immunohistochemistry for *Tp1*:H2BmCherry revealed that only ~50% of melanophores were marked by mCherry expression in wild-type fish (*Figure 5B*). To test whether this level of activity reflected signaling from cells of the xanthophore lineage, we examined *colony stimulating factor-1 receptor a (csf1ra)* homozygous mutants, which lack virtually all xanthophores and xanthoblasts (*Parichy et al., 2000*; *Patterson and Parichy, 2013*) (*Figure 5—figure supplement 2*), and found a marked reduction in the incidence of Tp1+ melanophores compared to wild-type siblings. To test for an airineme-dependence of such signaling, we examined airineme-inhibited fish expressing *aox5*: Tet:dnCdc42 and found a significant reduction in Tp1+ melanophores compared to dd-treated but non-transgenic siblings (*Figure 5B*).

If xanthoblast airinemes help transduce a Delta-Notch signal to melanophores important for stripe consolidation, then inhibition of Notch signaling generally, or within melanophores specifically, should cause defects in melanophore patterning similar to *aox5*:Tet:dnCdc42 and *dlc* mutant phenotypes. Indeed, global pharmacological inhibition of Notch signaling resulted in disruptions to melanophore stripe consolidation (*Figure 5—figure supplement 3A*). We therefore generated fish transgenic for a temporally inducible dominant negative Suppressor of Hairless [dnSu(H)] driven in the melanophore lineage with the *mitfa* promoter. *mitfa*:Tet:dnSu(H) reduced expression of Notch target gene *her6* but did not affect total numbers of melanophores or xanthophores that differentiated (*Figure 5—figure supplement 3B–D*). Similar to effects of dnCdc42-airineme inhibition, however, melanophores persisted ectopically in the interstripe (*Figure 5C,D*). Concordant effects of *aox5*:Tet:dnCdc42 and *mitfa*: Tet:dnSu(H) on melanophore patterning were

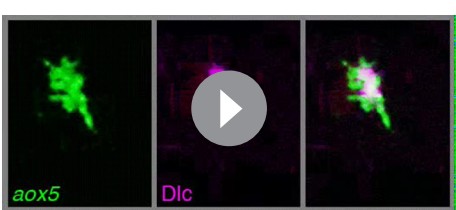

**Video 17.** Airineme vesicles harbored Dlc-mCherry. Left, *aox5*+ cell (green) extending airineme. Middle, *dlc*:Dlc-mCherry (magenta). Right, merge. 5 min interval, 170 min total.

also observed in regenerating fin (*Figure 5—figure supplement 4*).

We reasoned that if airinemes contribute to Delta-Notch signaling then such interactions could be revealed by synergistic effects of *aox5*:Tet:dnCdc42 and *mitfa*:Tet:dnSu(H) transgenes. Accordingly, we induced transgenes at very low levels that were insufficient to cause a phenotype when transgenes were present individually. When transgenes were combined, however, ectopic melanophores persisted in the interstripe (*Figure 5E*). Conversely, fish induced to express *aox5*:Tet: dnCdc42 at standard levels had significantly fewer ectopic melanophores when transgenic simultaneously for *mitfa*:NICD1a (mean ± SE melanophores: dnCdc42, 26.4 ± 1.1; dnCdc42 with NICD1a, 9.2 ± 1.5; $F_{1,8}$=84.5, p<0.001).

Because amniote melanocytes and most melanophores require the Kit (Kita) receptor tyrosine kinase for migration and survival (*Tan et al., 1990*; *Parichy et al., 1999*; *Rawls and Johnson, 2003*; *Wehrle-Haller, 2003*; *Budi et al., 2011*), we further speculated that airineme-dependent Notch signaling might act through Kita to promote melanophore clearance. Consistent with this idea, we found that melanophores of fish heterozygous for a *kita* null allele were sensitized to very low level induction of *mitfa*:Tet:dnSu(H) and *aox5*:Tet:dnCdc42 (*Figure 5—figure supplement 5A,B*). Abundances of *kita* transcript were also moderately reduced in melanophores, both in the absence of xanthophores and after inducing *mitfa*:Tet:dnSu(H) or *aox5*:Tet:dnCdc42 transgenes (*Figure 5—figure supplement 5C*). Indeed, melanophores of pearl danio, which migrate less than zebrafish (*Quigley et al., 2005*), exhibited similar reductions in Notch-responsive *her6* and *kita* transcripts (*Figure 5—figure supplement 5D*).

These observations support a model in which *aox5+* airinemes extended by xanthoblasts promote Notch signaling and Kita-dependent melanophore rearrangements during stripe consolidation in zebrafish.

## Evolutionary changes in xanthophore differentiation alter the potential for airineme signaling

To better understand how species differences in airineme production evolved, we re-examined pearl danio (*Figure 1A,C,E*), in which initially dispersed melanophores fail to migrate, resulting in a nearly uniform pattern that obscures latent (but genetically detectable) stripes (*Quigley et al., 2005*; *Mills et al., 2007*). Contributing to this phenotype is an early, widespread differentiation of xanthophores that likely attenuates the positional information available to melanophores, in contrast to zebrafish, in which a biased differentiation of xanthophores at the interstripe provides a cue for melanophore patterning (*Patterson et al., 2014*). In light of these prior findings, and the differentiation state dependence of airinemes in zebrafish (*Figure 3*), we hypothesized that pearl danios exhibit fewer *aox5+* airinemes owing to the precocious differentiation of their xanthophores. Consistent with this idea, experimental arrest of *aox5+* cell differentiation in pearl danio (TH–) increased airineme production, albeit not to levels observed in zebrafish (*Figure 6A*-left).

To test if species differences in airineme production reflect evolutionary changes that are autonomous or non-autonomous to the xanthophore lineage, we transplanted cells between zebrafish and pearl danio (*Parichy and Turner, 2003a*; *Quigley et al., 2004*). Zebrafish *aox5+* cells in pearl danio hosts extended airinemes at reduced frequencies, similar to that of pearl danio *aox5+* cells (*Figure 6A*-right; *Video 18*). We interpret this observation, as well as *aox5+* cell behaviors in reciprocal transplants, and adult pigment patterns of chimeras (*Figure 6—figure supplement 1*), as indicating species differences that are non-autonomous to the xanthophore lineage. These results are consistent with prior analyses that implicated *cis*-regulatory changes affecting environmentally produced xanthogenic Csf1 in the earlier and broader differentiation of xanthophores and altered melanophore pattern of pearl danio (*Patterson et al., 2014*). We therefore asked whether Csf1 alone could be responsible for species differences in airineme production. Consistent with this idea, overexpression of Csf1 in zebrafish resulted in a 22% reduction in airineme frequency within 24 hr and an 86% reduction after complete xanthophore differentiation, as in pearl (*Figure 6A*-right; *Video 19*).

Together, these results support the interpretation that infrequent airineme extension, as well as pattern differences in pearl danio as compared to zebrafish, arise at least in part from evolutionary changes in factors extrinsic to the xanthophore lineage, and identify Csf1 as an especially good candidate for mediating these effects.

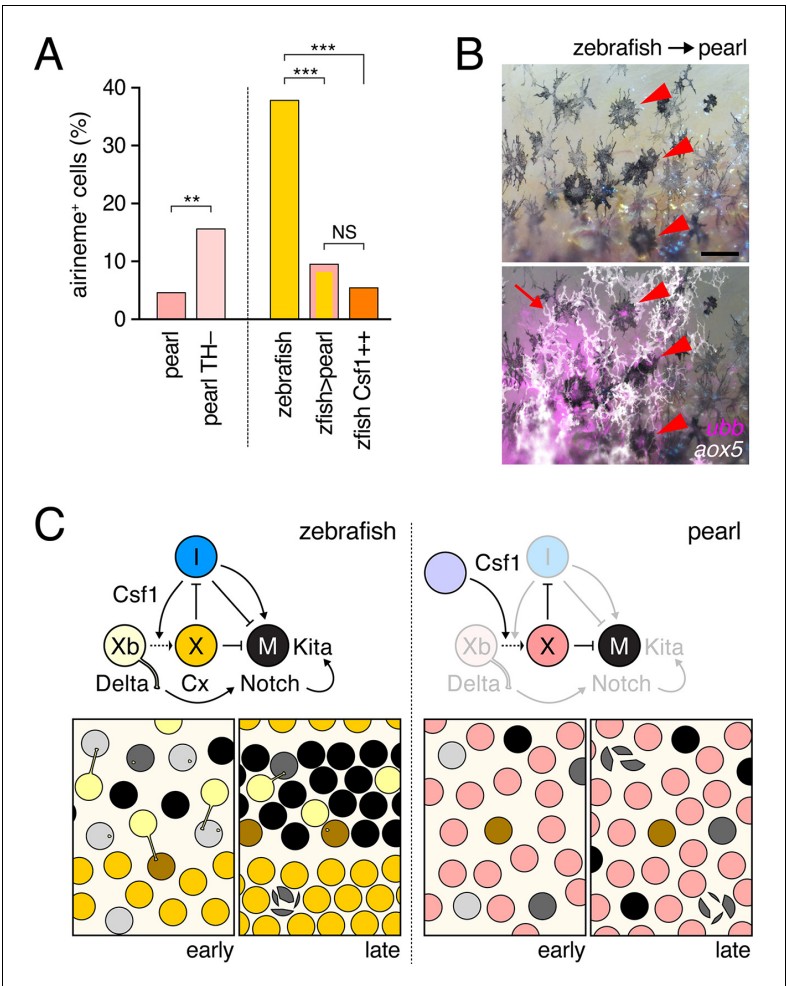

**Figure 6.** Factors extrinsic to the *aox5+* cells inhibit airineme production and signaling in pearl and models for airineme signaling and its evolution. (A) Left, Pearl *aox5+* cells extended more airinemes when differentiation-arrested (TH–; $\chi^2$=12.5, d.f.=1, p<0.0005, N=412 cells). Right, zebrafish *aox5+* cells transplanted to pearl, or receiving excess Csf1 in zebrafish, extended fewer airinemes than comparably staged *aox5+* cells in unmanipulated zebrafish ($\chi^2$=23.1, 22.1, d.f.=1, 1; p<0.0001, N=846 cells total). (B) In chimeras resulting from transplants of zebrafish donors (*aox5:membrane-GFP*, ubiquitous *ubb:mCherry* [**Mosimann et al., 2011**]) to pearl hosts, zebrafish *aox5+* cells (arrow) were typically intermingled with pearl melanophores, as well as *ubb* + zebrafish melanophores (arrowheads). (C) Working models for pigment cell interactions and pattern formation in zebrafish (left) and pearl danio (right). In zebrafish, xanthoblasts in stripe regions extend airinemes that signal to melanophores (Xb→M), promoting their clearance from the interstripe during stripe consolidation. Results of this study are consistent with interactions involving xanthoblast airineme dependent Delta (Dlc or possibly Dll4) activation of Notch signaling in melanophores, and the potentiation of Kita-dependent melanophore motility. Nevertheless, these data do not exclude roles for additional modes of Delta–Notch signaling, or the possibilities that airinemes transduce additional signals, or signals that cannot be distinguished from the Delta–Notch pathway using the experimental paradigms here employed. Analyses of mutant zebrafish further support roles for Cx41.8 in contributing to airineme-dependent communication, through modulation of target specificity or xanthophore lineage differentiation. In addition to xanthoblast–melanophore interactions, iridophores have attractive and repulsive effects on melanophores (I→M; I⊣M) (**Frohnhofer et al., 2013**; **Patterson and Parichy, 2013**) and express Csf1, promoting the differentiation of xanthophores (Xb→X) at the interstripe (**Patterson and Parichy, 2013**). Differentiated xanthophores repel melanophores (X⊣M) during normal development (**Nakamasu et al., 2009**) and are capable of repressing iridophore organization (X⊣I) [for details, see: (**Patterson et al., 2014**)]. In pearl danio, Csf1 is expressed at elevated levels by cells other than iridophores and this drives earlier and broader xanthophore differentiation than in zebrafish (**Patterson et al., 2014**). Precocious, widespread differentiation of xanthophores likely limits directional cues available to melanophores while simultaneously curtailing the potential for airineme signaling, as airineme competent xanthoblasts are depleted. Tissue contexts (lower panels) also show eventual death of some melanophores remaining in the interstripe in zebrafish (**Parichy et al., 2000**; **Parichy and Turner, 2003b**) and the higher overall incidence of melanophore death in pearl danio (**Quigley et al., 2005**); iridophores are omitted for clarity. Scale bar: 50 µm (B).

The following figure supplement is available for figure 6:

**Figure supplement 1.** Interspecific chimeras reveal non-autonomous effects on pattern and *aox5+* cell behaviors.

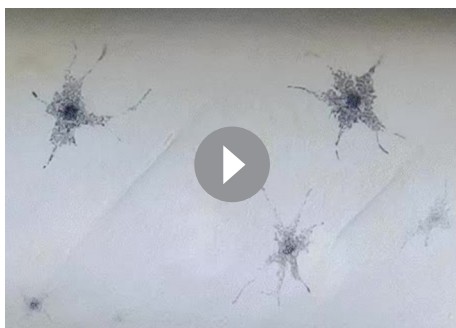

**Video 18.** Zebrafish *aox5*+ cells transplanted to pearl hosts behaved like pearl *aox5*+ cells. Zebrafish-derived cells expressed *ubb*:mCherry (magenta) and zebrafish cells of the xanthophore lineage were co-labeled with *aox5*:membrane-GFP (green). Airinemes were very rarely extended by zebrafish *aox5*+ cells in the pearl background; a single such example is shown (arrow). Also visible are *ubb*+ muscle cells deeper in the fish. Spots of green were autofluorescing host xanthophores, and melanophores shown in bright field were of host origin. 5 min interval, 370 min total.

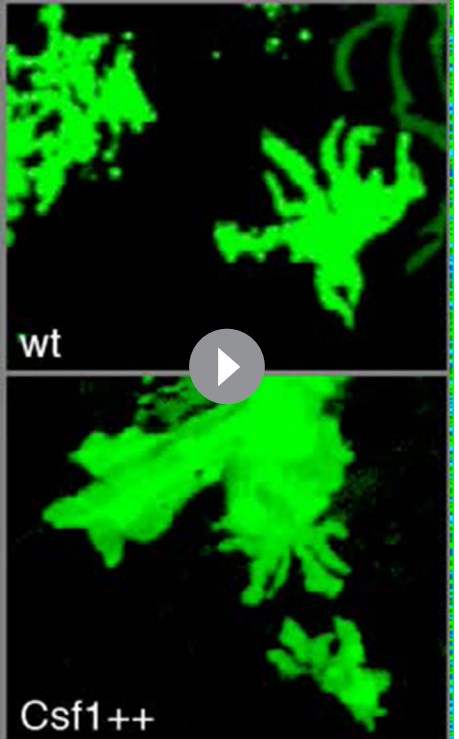

**Video 19.** Transgenic expression of *hsp70l*:Csf1a blocked airineme production in zebrafish. Upper, an *aox5*+ cell extends an airineme (arrow) in a heat-shocked, non-transgenic control fish, that stabilized on an adjacent melanophore (magenta). Additional *aox5*+ vesicles were evident as well. Lower, in heat-shocked transgenic fish in which differentiation had been forced by Csf1a overexpression, airineme and vesicle production were not evident. 5 min interval, 810 min total.

## Discussion

Our findings suggest a revised model for how adult stripes form in zebrafish, and link evolutionary genetic modifications to morphogenetic behaviors occurring during the development of a very different, naturally occurring phenotype, the nearly uniform pattern of pearl danio (*Figure 6C*). More generally, our approaches have identified a novel mode of long distance cellular communication, and provide fresh insights into evolutionary changes in morphogenesis during pigment pattern development.

A particularly striking finding is the role in zebrafish stripe consolidation played by long-distance cellular projections—airinemes—extended by xanthoblasts in prospective stripe regions. Genetic analyses have shown that interactions between melanophore and xanthophore lineages are important for stripe formation and maintenance in zebrafish (*Parichy et al., 2000*; *Maderspacher and Nusslein-Volhard, 2003*; *Parichy and Turner, 2003a*), whereas laser ablation and in vitro studies suggest that such interactions can occur over both short- and long-range (*Takahashi and Kondo, 2008*; *Nakamasu et al., 2009*; *Yamanaka and Kondo, 2014*). By examining the behaviors of differentiated and undifferentiated cells in their native tissue environment using membrane-targeted fluorophores and high resolution time-lapse imaging, we identified novel interactions, as well as unanticipated differentiation-state dependencies to these interactions. Our data suggest a model in which xanthoblasts sparsely populating prospective stripe regions, but not xanthophores of the interstripe, extend Delta+ airinemes that contact melanophores, promoting Notch signaling and melanophore consolidation into stripes (*Figure 6C*-left). Airineme-delivered signals could provide directional information, or could simply promote motility, with directionality provided some other way. Indeed, differentiated xanthophores repel melanophores (*Takahashi and Kondo, 2008*; *Yamanaka and Kondo, 2014*), and these effects seem likely to synergize with airineme-mediated interactions to promote stripe consolidation. Additional approaches to testing this model (independent of dnCdc42 activities) should become feasible as mechanisms of airineme production and targeting are elucidated. The adult pattern defects we observed also illustrate how failures in even the subtle initial movements of adult melanophores during interstripe clearance and stripe

consolidation (*Parichy et al., 2000*; *Parichy and Turner, 2003a*; *Takahashi and Kondo, 2008*) are amplified by the considerable somatic growth that occurs during pigment pattern formation (*Parichy et al., 2009*).

Our results have implications for theoretical models of pattern formation. Short-and long-range interactions between melanophore and xanthophore lineages are concordant with a Turing model (*Meinhardt and Gierer, 2000*; *Nakamasu et al., 2009*; *Kondo and Miura, 2010*; *Watanabe and Kondo, 2015*). Although such models classically envisaged diffusible substances, the same outcomes could, in principle, be achieved in many ways (*Hamada et al., 2014*). Our findings are likely consistent with the current formulation of a Turing model as applied to zebrafish pigment pattern. Nevertheless, our discovery that airineme production and targeting depend on precise states of differentiation reveals a previously unappreciated complexity to these signaling dynamics. Moreover, the nature of interactions seems likely to differ across stages: in adult fish, melanophore processes reach towards interstripe xanthophores, and may allow for long-range Delta–Notch trophic support (*Hamada et al., 2014*); yet, we did not observe long, stable melanophore processes during pattern development (e.g., *Video 3*). Expansions of existing theory to incorporate differentiation state heterogeneity, stage-specificity of interactions, and additional cell types would seem a valuable endeavor.

Our study adds to the known repertoire of specialized projections by which cells communicate (*Miller et al., 1995*; *Ramirez-Weber and Kornberg, 1999*; *Caneparo et al., 2011*; *McKinney et al., 2011*; *Bischoff et al., 2013*; *Massarwa and Niswander, 2013*; *Sanders et al., 2013*; *Gradilla et al., 2014*; *Luz et al., 2014*; *Roy et al., 2014*; *Stanganello et al., 2015*). In contrast to actin-based projections (e.g., cytonemes), airinemes required both microfilaments and microtubules, and delivered large, membrane-bound vesicles that persisted on target cells up to several cell diameters away. By analogy with neuronal dendrites, it seems plausible that airinemes participate not only in 'forward' signaling, shown here, but also 'reverse' signaling that could allow cells to sense their environment and regulate differentiation or morphogenesis accordingly (though we could not detect differences in xanthophore differentiation after increasing melanophore abundance or blocking airineme production; *Figure 1—figure supplement 2B*; *Figure 4—figure supplement 2E*). Cellular mechanisms of airinene production and targeting, as well as additional modalities for airineme signaling, are currently being investigated. It will be interesting to discover whether airinemes are used for signaling in other tissue contexts as well.

Finally, interspecific transplants and transgenic manipulations in this study provide new insights into the nearly uniform pattern of pearl danio, and together suggest a model in which enhanced Csf1 expression drives precocious, widespread xanthophore differentiation, limiting positional information available to melanophores (*Patterson et al., 2014*), as well as airineme production and the potential for melanophore migration and stripe consolidation (*Figure 6C*—right). These findings do not exclude roles for additional factors; e.g., melanophore-autonomous differences, or changes in iridophore patterning that may or may not themselves be xanthophore-dependent. Nevertheless, by focusing at high resolution on cellular behaviors our study has provided insights into the evolution of an alternative pattern state that could not have been anticipated from analyses of genetic variation and gene regulatory differences alone. These findings illustrate how an iterative approach with model organisms and closely related species can provide insights into mechanisms by which evolutionary changes in gene activity are translated through morphogenetic behaviors into species differences in form.

## Materials and methods

### Staging, rearing and stocks

Staging followed (*Parichy et al., 2009*) and fish were maintained at 28.5°C, 16:8 L:D. Zebrafish were wild-type AB[wp] or its derivative WT(ABb) and *kita*[b5] (*Parichy et al., 1999*), *csf1ra*[j4e1] (*Parichy et al., 2000*), *mitfa*[w2] (*Lister et al., 1999*), *ltk*[j9s1] (*Lopes et al., 2008*), *dlc*[b663] (from S. Amacher), hyperthyroid *opallus*[b1071] (*tshr*[D632Y]), Tg(tg:nVenus-2a-nfnB)[wp.rt8], Tg(aox5:palmEGFP)[wp.rt22], Tg(tyrp1b:palm-mCherry)[wp.rt11] (*McMenamin et al., 2014*),Tg(hsp70:kitga)[wp.rt2], Tg(hsp70:csf1a-DrIRES-nlsCFP)[wp.rt4] (*Patterson and Parichy, 2013*),Tg(Tp1:H2b-mCherry) (from N. Ninov, D. Stainer (*Ninov et al., 2012*)], Tg(aox5:TetGBD-TREtight:nVenus-2a-dnCdc42)[wp.rt19] ['*aox5:Tet:dnCdc42*'], Tg[mitfa:TetGBD-TREtight:nVenus-2a-dnSU(H)]*[wp.rt20]* ['*mitfa:Tet:dnSu(H)*'], Tg(mitfa:nVenus-2a-NICD)[wp.rt21].

Pearl danio *D*.aff. *albolineatus* (*Quigley et al., 2005*) were wild type, *Tg(tg:nVenus-2a-nfnB)*[wp.at3] (*McMenamin et al., 2014*) and *Tg(aox5:palmEGFP)*[wp.at4].

## Transgenesis and transgenic line production

Cell-type and temporally specific expression of human dnCdc42 (from J. Wallingford) or dnSu(H) (from D. Stainier) used doxycycline and dexamethasone (dd) inducible TetON, TetGBD (*Knopf et al., 2010*) upstream of TREtight (*Patterson and Parichy, 2013*). Effectors were linked by 2a sequences to nuclear-localizing Venus (nVenus) and driven by 8 kb *aox5* and 2.2 kb *mitfa* promoters in xanthophore and melanophore lineages. Inductions were performed on F0 mosaic fish and non-mosaic stable lines with similar results. Constitutive Notch activation used the intracellular domain of *notch1a* linked by 2a sequence to nVenus, and driven by the *mitfa* promoter. To visualize DeltaC localization under native regulatory elements, we inserted mCherry C-terminally using BAC CH73-208H7 with 34 kb 5' and 69 kb 3' to the open reading frame; DlC-mCherry functionality was verified in mosaic F0 fish by rescue of an embryonic segmentation defect in 47% of *dlc* loss of function embryos. Actin was examined with mCherry fused to the utrophin calponin homology domain [Addgene #26740; (*Burkel et al., 2007*)] or LifeAct-mKate [from M. Barna]. Microtubule labeling used subclones from Tuba1-mCherry [(Addgene #49149; (*Friedman et al., 2010*)] or EB3-EGFP (Michael Davidson, Addgene #56431).

## Drug treatments and heat shock experiments

Fish were treated with drugs during dark cycles then changed to water without drugs at light cycles. TetGBD transgenics or non-transgenic siblings were given 25 nM doxycycline and 50 µM dexamethasone (dd; Sigma-Aldrich, St. Louis, MO) from 6.5 SSL. Tests for interactions with transgenes used 6 nM and 12.5 µM dd. Notch inhibitor LY411575 (3 µM) and Cdc42 inhibitor ML141 (1 µM) were prepared in DMSO and fish treated with drug or vehicle from 6.0 SSL. For acute drug administrations, ML141 (2 µM), blebbistatin (10 µM) or nocodazole (3 µg/ml) were added to ex vivoculture medium (below) 30 min prior to time-lapse imaging. TH– fish were generated by ablating thyroid follicles of *Tg(tg:nVenus-2a-nfnB)* at 4 d (*McMenamin et al., 2014*). For heat shock, *Tg(hsp70:kitlga)* and *Tg (hsp70:csf1a-DrIRES-nCFP)* larvae were exposed 1 hr to 38°C twice daily between 6.5–7.0 SSL and 7.5 SSL.

## Time-lapse and still imaging

Ex vivo imaging of pigment cells in their native tissue environment followed (*Budi et al., 2011*; *Eom et al., 2012*), with images acquired at 5-min intervals for 18 hr at 10x using an Evolve (Photometrics, Tucson, AZ) camera mounted on a Zeiss Observer Z1 inverted microscope with CSU-X1 spinning disk (Yokogawa, Tokyo, Japan). Bright-field images were taken before and after imaging. Larvae were 7.5 SSL except where indicated. For analyses of *aox5+* airinemes across stages, genetic backgrounds or both, labeled cells were examined in 79 wild-type zebrafish and 40 wild-type pearl; 4 *opallus*, 4 *ltk*, 10 *mitfa*, and 21 *cx41.8* mutants; 11 *Tg(hsp70:csf1a-DrIRES-nlsCFP)*, 4 *Tg (hsp70:kitlga)*, 5 *Tg(aox5:Tet:dnCdc42)*, 7 *Tg(tg:nVenus-2a-nfnB)* zebrafish and 13 *Tg(tg:nVenus-2a-nfnB)* pearl; 8 ML141-treated, 4 blebbistatin-treated, 4 nocodazole-treated, 3 DMSO control zebrafish. Analyses of *tyrp1b+* airineme incidence used 56 wild-type zebrafish. Images in *Figures 1B*, *5D* and Video 6 were taken with Zeiss (Jena, Germany) LSM880 or LSM800 scanning laser confocal microscopes with Airyscan detectors.

## Cell counts and distributions

Pigment granules were contracted with epinephrine to facilitate counts and assessment of cell centroids using the Cell Counter plugin of ImageJ (NIH). To assess melanophore pattern, each cell was assigned to one of 20 bins from dorsal (0.0) to ventral (1.0), and total numbers of melanophores within each bin were averaged across individuals. Positions of interstripes were indicated by melanophore-free troughs in melanophore distributions of controls (positions 0.40, 0.45, 0.50); numbers of melanophores at these locations were compared for assessment of interstripe melanophore persistence.

## RT-PCR

For all quantitative RT-PCR analyses, melanophores were isolated in 2.5 mg/ml trypsin in phosphate buffered saline (PBS) for 10 min at 36°C. Tissues were rinsed with PBS, and incubated at 28°C in 1 mg/ml collagenase I, 0.1 mg/ml DNase I, 0.1 mg/ml trypsin inhibitor in PBS while shaking. Suspension solutions were filtered, centrifuged 30x *g* for 10 min at 4°C in 50% Percoll density gradients to precipitate melanophores, assessed for purity at high magnification, then extracted for RNA with an RNAqueous-Micro kit (Thermo Fisher, Waltham, MA). cDNA was synthesized with the iScript cDNA Synthesis Kit (Bio-Rad, Hercules, CA). Quantitative PCR was performed on an Applied Biosystems StepOne Plus using custom TaqMan probes and primers spanning intron–exon boundaries and targeting regions of sequence identity between zebrafish and pearl (*her6*, AIRSA5P; *kita*, AILJKAN; *rpl13a*, AI5H3GZ). Quantitative PCRs were run with at least triplicate biological and technical replication.

For non-quantitative RT-PCR, 100–150 melanophores or xanthophores were picked by micromanipulator from dissociated cells. Amplifications were 40 cycles (*actb1*, *dlc*, *csf1ra*, *pmela*, *cx41.8*, *cx39.4*) or 45 cycles (*dll4*) at 94°C, 30 s; 62°C, 20 s; 72°C, 20 s. Absence of cross-contamination was verified by expression of *pmela* or *csf1r*, specific to melanophores and xanthophores, respectively (*Kelsh et al., 2000*; *Parichy et al., 2000*). *actb1*: 5′- ACTGGGATGACATGGAGAAGAT3′, 5′- GTGTT-GAAGGTCTCGAACATGA-3′; *csf1r*: 5′-CAGAGTGACGTCTGGTCTTACG-3′, 5′-GGACATCTGATAG-CCACACTTG-3′; *cx39.4*: 5′- GACAGTCTTCCAAGCTACTCAA-3′, 5′-GGTGCTCTGCTTCTCAAACATA-3′; *cx41.8*: 5′-ACATCCGGTCAACTGCTACAT-3′, 5′- TTGTATCC-GTGCACATACTTCC-3′; *dlc*: 5′-CGGGAATCGTCTCTTTGATAAT-3′, 5′-CTCACCGATAGCGAGTC-TTCTT-3′; *dll4*: 5′- GCACTCACCTTACTCGGATCTA-3′, 5′-CACTTTGAACATCCTGAGACCA-3′ *pmela*: 5′-CTCGGAGTTCTGTTTTTCGTTT3′, 5′- AAGGTACTGCGCTTATTCCTGA-3′.

## Immunohistochemistry

To verify transgene expression, 7.5-SSL larvae were fixed in 4% paraformaldehyde, embedded in OCT (Tissue-Tek, VWR, Radnor, PA), and cryosectioned at 20 μm. Sections were washed with 0.3% Triton X-100 in PBS (PBSTX), blocked, then incubated at 4°C overnight with primary antibodies. Primary antibodies were mouse anti-PAX7 (Developmental Studies Hybridoma Bank) for the xanthophore lineage (*Minchin and Hughes, 2008*) and rabbit anti-GFP (ThermoFisher) for Venus. Notch signaling reporter *Tg(TP1:H2B-mCherry)* was visualized with rat anti-mCherry (ThermoFisher).

## Cell transplantation

Chimeric fish were generated by transplanting cells at blastula stages then rearing embryos reared through adult pigment pattern formation (*Parichy and Turner, 2003a*; *Quigley et al., 2004*) using fish that were transgenic for ubiquitous *ubb:mCherry* or *actb1:EGFP*, xanthophore-lineage *aox5: membrane-GFP,* or both.

## Statistical analyses

Analyses were performed with JMP 8.0 (SAS Institute, Cary, NC). Frequency data for behaviors of individual cells or projections were assessed by single or multiple factor maximum likelihood. Continuous data were evaluated by *t*-test or analyses of variance, using *ln*-transformation in some instances to correct residuals to normality and homoscedasticity. Post hoc means were compared by Tukey-Kramer HSD.

## Acknowledgements

For assistance or discussions we thank Inseok Hwang, Jiae Lee, Tracy Larson, Thao Pham, and Jay Parrish.

## Additional information

### Funding

| Funder | Grant reference number | Author |
|---|---|---|
| National Institute of General Medical Sciences | NIH R01 GM096906 | David M Parichy |
| National Institute of General Medical Sciences | NIH R01 GM111233 | David M Parichy |
| National Institute of Child Health and Human Development | T32 HD007183 | Emily J Bain |

The funders had no role in study design, data collection and interpretation, or the decision to submit the work for publication.

### Author contributions

DSE, EJB, Conception and design, Acquisition of data, Analysis and interpretation of data, Drafting or revising the article; LBP, Acquisition of data, Analysis and interpretation of data, Drafting or revising the article; MEG, Acquisition of data; DMP, Conception and design, Analysis and interpretation of data, Drafting or revising the article

### Author ORCIDs

Emily J Bain, http://orcid.org/0000-0003-0428-4357

### Ethics

Animal experimentation: This study was performed in strict accordance with the recommendations in the Guide for the Care and Use of Laboratory Animals of the National Institutes of Health. All animals were handled according to approved institutional animal care and use committee (IACUC) protocol (#4094) of the University of Washington.

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
