## [Decision Letter]

Thank you for submitting your work entitled "Long-distance communication by specialized cellular projections during pigment pattern development and evolution" for consideration by *eLife*. Your article has been reviewed by three peer reviewers, and the evaluation has been overseen by a Reviewing Editor and Diethard Tautz as the Senior Editor.

The reviewers have discussed the reviews with one another and the Reviewing Editor has drafted this decision to help you prepare a revised submission.

Summary:

The consensus view of the reviewers is that this study presents a series of genetic and experimental perturbations in zebrafish and pearl danio that provides new insight into the morphogenetic and developmental basis of periodic color pattern. They provide dynamic live imaging of pigment cells in developing zebrafish, leading to the discovery that (mostly) xanthophores exhibit an unusual type of cytoplasmic projection that appears to be involved in stripe consolidation. The authors characterize these projections, which they term airinemes, with regard to their development and composition. Transgenic experiments provide evidence that airinemes are an essential component of a xanthoblast to melanophore signal that promotes melanophore movement and stripe consolidation possibly via a δ-Notch signal. There are also associated changes in melanophore Kit expression. They also show that the pearl danio, which lacks melanophore stripes, also lacks airinemes.

Overall, the work is carefully and critically described, and will be of widespread interest and significance to developmental and evolutionary biologists. However, there is a general concern that the presentation or description in the text needs to be improved with regard to strength of evidence relative to interpretation of the data. The data in the first half of the paper is excellent but the second part of the work is suggestive not proven. This issue can primarily be addressed by changes to the presentation, and, possibly, an additional straightforward experiment.

To aid the authors in considering how to revise the text we are communicating the major points of individual reviewer opinions below.

1) The first half of the manuscript (identification, cell biologic and developmental characterization of airinemes) is strong, but the second half (Delta-notch, kita expression, differentiation dependence, xanthophore non-autonomy) is less so. The Delta-Notch experiments are an important addition to the manuscript, but the conclusions regarding a requirement for Delta-Notch signaling are not as strong as the conclusions regarding a requirement for airinemes. The existing evidence is enough to speculate about the involvement of Notch but in the absence of additional data it is essential that changes are made accordingly to the Discussion and the model in Figure 8 that take into account the relative strength of the conclusions.

a) The DLC-mCherry fusion protein appears to be found throughout xanthophores (cell body, projections, vesicles), which is not quite the smoking gun that one might have hoped for. What about Dll4? (Is there any genetic or molecular biologic information that makes dll4 a less likely candidate?) The Notch reporter experiment is interesting but it would be helpful to have some more context. Is the reporter expressed in any other pigment cells? Does the melanophore signal described by the authors stand out relative to other evidence of Notch signaling during pigment cell migration and stripe consolidation. Finally, while the result of the double mutant (dnCdc42; dnSu(H); low levels of dd) is interesting, it doesn't quite reach the level of rigor one would hope for in an epistasis analysis. A stronger experiment would be to show that increased Notch signaling in melanophores could rescue the phenotype of a xanthophore dnCdc42 transgene. Showing that constitutive notch (mitfa>NICD) suppresses aox5>dnCDC42 phenotype of interstripe melanocyte would be more direct (and more rigorous) than the experiments showing synergistic genetic interaction between notch pathway (dnSuH) in mels and CDC42 promoted airinemes in xans. Together, these two experiments would make the point of airinemes acting through Notch rather compelling. This seems like a fairly simple experiment with the available fish stocks already used in study.

b) The melanophore kita expression results are suggestive but not developed. Since the phenotype is a failure of mel migration, and kit is required for migration, it is justified to explore the role of kit in the phenotype. The results shown – stronger phenotype of genetic interactions (dnSuH, dnCDC42) in kit(-/+) than in kit(+/+), together with the evidence that Notch signalling controls kit expression is sufficient to tell us there is a role. The key question is to look for how kit is responsive to Notch signalling. As it stands in the paper kita is a candidate gene/pathway, the level of reduction is not very substantial, and it's not clear exactly how melanophores were purified/enriched (the Methods describes both Percoll purification and micromanipulation) or the degree of enrichment. If RNA-seq information is available on purified melanophores with and without airineme contact, that would add a lot. This data is not absolutely necessary for the manuscript to move forward but conclusions need to at least be toned down.

c) The TH and the transplantation experiments are similar to the Delta-Notch experiments – the results are interesting but not compelling.

2) It is surprising that this type of cell protrusion has not been described before. The authors themselves, and many other groups as well, have used live imaging of Xanthophores but never reported such a phenomena. The authors should comment in the paper why this is the case. Have they been simply overlooked so far or are technical advances now available that have led to this discovery?

3) How sure can the authors be that these structures are indeed different from previously identified cytonemes. There are cases of cytonemes that stain positive for tubulin. The paper also exaggerates a bit the absolute and unique roles of these "airinemes" in the text, while a closer look into the graphs and figures shows that such absolute statements are not correct. For example: "Thus, exuberant production of long, fast, and vesicle-containing projections was UNIQUE to the zebrafish xanthophore lineage…" Figure 1 however, shows protrusions both in melanophores and in danio pearl xanthophores – simply in lower numbers. Also, the difference seems to be restricted to stage 7-8. While this obviously could be enough to have an effect, a more veridic description of the observations throughout the text would be helpful.

---

## [Author Response]

*To aid the authors in considering how to revise the text we are communicating the major points of individual reviewer opinions below.*

*1) The first half of the manuscript (identification, cell biologic and developmental characterization of airinemes) is strong, but the second half (Delta-notch, kita expression, differentiation dependence, xanthophore non-autonomy) is less so. The Delta-Notch experiments are an important addition to the manuscript, but the conclusions regarding a requirement for Delta-Notch signaling are not as strong as the conclusions regarding a requirement for airinemes. The existing evidence is enough to speculate about the involvement of Notch but in the absence of additional data it is essential that changes are made accordingly to the Discussion and the model in Figure 8 that take into account the relative strength of the conclusions.*

Thank you for raising this issue. We do believe that our several analyses, including a significant reduction in Notch signaling reporter activity in melanophores that results from airineme inhibition, as well as the phenotypes of the *dlc* mutant and several transgenic lines singly and in combination with one another, together support the notion that Delta-Notch signaling is important for stripe patterning and that airinemes very likely contribute to the transduction of such signals. Nevertheless, our original presentation was not intended to imply that airinemes are the only mechanism by which Delta–Notch signaling occurs between melanophore and xanthophore lineages (e.g., our own prior analyses in collaboration with Shigeru Kondo’s group in Hamada et al. 2014) or that airinemes exclusively transduce such a signal. The working model we presented and its original description were merely intended to represent our best current understanding of the system according to the evidence currently available. To minimize any potential for misunderstanding, we have made these points more explicit in the discussion and legend of Figure 6 (corresponding to original Figure 8).

*a) The DLC-mCherry fusion protein appears to be found throughout xanthophores (cell body, projections, vesicles), which is not quite the smoking gun that one might have hoped for.*

Although Dlc-mCherry is present widely over xanthoblast cell membranes, we do observe especially abundant concentrations associated with the blebs from which airinemes originate and we have replaced the original images to better illustrate this point. Nevertheless we did not intend to imply that Dlc becomes specifically concentrated at blebs or in airineme vesicles, merely that such structures could harbor the protein if it happens to already be at the cell surface, and our observations support this view. Indeed, we expect there are many ways in which melanophores could “obtain” a Delta signal from the xanthophore lineage, including the extension of processes from melanophores to xanthophore cell bodies as appears to happen at later stages (Hamada et al. 2014). Although we are interested in whether specific proteins are actively localized at sites of airineme emergence, and we are engaged in work to address this point, we feel this issue is beyond the scope of the present study as our interpretations do not require such a mechanism.

What about Dll4? (Is there any genetic or molecular biologic information that makes dll4 a less likely candidate?)

We suspect Dll4 could well be involved, too. Unfortunately, we have not been able to find a BAC clone containing Dll4 that is suitable for generating a fusion protein to test its localization in the context of native regulatory elements; so we have yet to pursue this candidate further. Nevertheless, our original intent with analyses of Dlc was merely to illustrate that at least one partner in Delta–Notch signaling can be found in airineme extensions and it was for this reason that we referred to potential interactions as involving generic “Delta” signals in the remainder of the manuscript, rather than specific “Dlc” signals. We have noted in the text that technical limitations have hampered analyses of Dll4.

*The Notch reporter experiment is interesting but it would be helpful to have some more context. Is the reporter expressed in any other pigment cells? Does the melanophore signal described by the authors stand out relative to other evidence of Notch signaling during pigment cell migration and stripe consolidation.*

We added more information about Notch reporter expression in the legend to Figure 5 and Notch pathway gene expression in the legend to Figure 5—figure supplement 1. In short, we observe plenty of Notch gene expression and Notch signaling in other tissues, as would be expected given the diversity of Delta–Notch functions during development, but in the hypodermis we detect Tp1 only in melanophores. Ours is the first study to explicitly address levels of Notch signaling within melanophores so we do not know if the levels we observe are greater or less than what might be present at other stages of development.

Finally, while the result of the double mutant (dnCdc42; dnSu(H); low levels of dd) is interesting, it doesn't quite reach the level of rigor one would hope for in an epistasis analysis. A stronger experiment would be to show that increased Notch signaling in melanophores could rescue the phenotype of a xanthophore dnCdc42 transgene. Showing that constitutive notch (mitfa>NICD) suppresses aox5>dnCDC42 phenotype of interstripe melanocyte would be more direct (and more rigorous) than the experiments showing synergistic genetic interaction between notch pathway (dnSuH) in mels and CDC42 promoted airinemes in xans. Together, these two experiments would make the point of airinemes acting through Notch rather compelling. This seems like a fairly simple experiment with the available fish stocks already used in study.

We added these data on *aox5*:dnCdc42 phenotypes with and without *mitfa*:NICD in the background; a significant reduction in ectopic melanophores when Notch activity is restored autonomously to melanophores supports our original interpretation.

*b) The melanophore kita expression results are suggestive but not developed. Since the phenotype is a failure of mel migration, and kit is required for migration, it is justified to explore the role of kit in the phenotype. The results shown* – *stronger phenotype of genetic interactions (dnSuH, dnCDC42) in kit(-/+) than in kit(+/+), together with the evidence that Notch signalling controls kit expression is sufficient to tell us there is a role. The key question is to look for how kit is responsive to Notch signalling. As it stands in the paper kita is a candidate gene/pathway, the level of reduction is not very substantial, and it's not clear exactly how melanophores were purified/enriched (the Methods describes both Percoll purification and micromanipulation) or the degree of enrichment. If RNA-seq information is available on purified melanophores with and without airineme contact, that would add a lot. This data is not absolutely necessary for the manuscript to move forward but conclusions need to at least be toned down.*

Our intent with analyses of *kita* was simply to assess whether the best known pathway promoting melanophore motility might be affected by alterations in airineme behaviors and Notch signaling. On-going studies are working to assess transcriptomic and other changes associated with airineme signaling but these analyses will require additional months or years of effort. In the meantime we have clarified that alterations in *kita* expression are moderate, as we would anticipate given that many factors likely contribute to activity of this pathway. We have also moved analyses of *kita* to supplementary materials to better reflect our own view of their relative import.

In the Methods we have further clarified that Percoll gradients were used for qPCR and cell picking by micromanipulator for non-quantitive RT-PCR. Purities of samples were assessed by microscopic examination at high magnification. Though we cannot absolutely exclude the possibility of some contaminating cells, the genes we have studied are already known to be expressed by melanophores from published analyses (and our own unpublished RNA-Seq studies on small numbers of individually picked cells) and we anticipate that, given the cell-type specificity of our manipulations, contaminating cells that might express the same genes would tend to reduce the sensitivity of our assays, biasing our results towards lower rather than higher levels of significance.

*c) The TH and the transplantation experiments are similar to the Delta-Notch experiments – the results are interesting but not compelling.*

Our only intent with TH and transplantation experiments were to use the former as a way to manipulate xanthophore differentiation and the latter to assess the cell autonomy of species differences in airineme production and pigment pattern. The results of these analyses were straightforward so it is not clear precisely how we would make them more compelling. We are indeed studying roles for TH in the xanthophore lineage more generally, and we are looking into additional factors that influence pattern differences between species, but these seem beyond the scope of the present study. Given concerns about these experiments, however, we have streamlined presentation of these analyses to ensure they are as concise as possible.

*3) It is surprising that this type of cell protrusion has not been described before. The authors themselves, and many other groups as well, have used live imaging of Xanthophores but never reported such a phenomena. The authors should comment in the paper why this is the case. Have they been simply overlooked so far or are technical advances now available that have led to this discovery?*

Our own previous live imaging focused on the behaviors of melanophore precursors using cytosolic reporters and fluorescence wide-field time-lapse imaging; other analyses have focused on behaviors of fully differentiated pigment cells using bright field imaging. To our knowledge, this is the first time that membrane targeted fluorophores and laser spinning disk imaging have been used to visualize dynamics of danio pigment cells and their precursors in the tissue environment. We have added a note to this effect in the Discussion.

*4) How sure can the authors be that these structures are indeed different from previously identified cytonemes. There are cases of cytonemes that stain positive for tubulin.*

Our reading of the literature suggests that projections of this size, with meandering trajectories, an actin and tubulin based cytoskeleton and very large vesicles that are delivered to other cells, are unique, or at the very least not widely known or previously named. We have also made informal enquiries with others studying long-range cellular communication, as well as prominent neuroanatomists; none have indicated that these projections resemble ones they have observed or heard about. Indeed, objections have been raised to our past use of the term ‘cytoneme’ in describing these projections in the context of seminar presentations, etc., precisely for the reasons listed above. We have therefore found it convenient, and hopefully less confusing, to name the structures rather than referring to them as “cytonemes”, “fast pigment cell projections”, “vesicle containing processes” etc. Regarding etymology, we could not pass up a once-in-a-career opportunity to reference Iris and *The Iliad*, and to validate the importance of a liberal arts education. We considered naming the processes “irinemes,” but we were concerned this might evoke “iridophores”, which are the one class of pigment cell that seems, so far, to lack them. Nevertheless, we believe the reference to fast message delivery is appropriate given our findings (of course “-neme” or “thread” honors the exciting biology of cytonemes and their discoverers). Finally, upon weighing the merits of several alternatives, we opted to hopefully avoid confusion with iridophores by modifying the name to “airineme,” which has the added advantage of honoring Sir George Airy’s contributions to optics.

*The paper also exaggerates a bit the absolute and unique roles of these "airinemes" in the text, while a closer look into the graphs and figures shows that such absolute statements are not correct. For example: "Thus, exuberant production of long, fast, and vesicle-containing projections was UNIQUE to the zebrafish xanthophore lineage*…*" Figure 1 however, shows protrusions both in melanophores and in danio pearl xanthophores* – *simply in lower numbers. Also, the difference seems to be restricted to stage 7-8. While this obviously could be enough to have an effect, a more veridic description of the observations throughout the text would be helpful.*

Regarding our descriptions we have always taken great care to accurately portray the data presented in all plots and images, as well as the totality of our observations overall. Although the veracity of our descriptions was not intentionally lacking, we acknowledge that some statements may have been open to multiple interpretations. We have therefore made the following changes:

Original: “Thus, exuberant production of long, fast, and vesicle-containing projections was unique to the zebrafish xanthophore lineage and occurred during stages of stripe formation.”

Revised: “Thus, long, fast, and vesicle-containing projections were produced exuberantly by cells of the zebrafish xanthophore lineage during stages of stripe formation, but were not common to other pigment cell classes in this species or to cells of the xanthophore lineage in pearl danio.”

Original: “To test if airineme production depended on differentiation state we exploited the thyroid hormone (TH) dependence of xanthophore differentiation”

Revised: “To test experimentally if airineme production was contingent upon differentiation state, we exploited the thyroid hormone (TH) dependence of xanthophore differentiation”

Original: “*kita* transcript abundances were likewise reduced in melanophores upon inhibition of Notch signaling [*mitfa:Tet:dnSu(H)*] or airineme production [*aox5:Tet:dnCdc42*] (Figure 6).”

Revised: “Abundances of *kita* transcript were also moderately reduced in melanophores, both in the absence of xanthophores and after inducing *mitfa*:Tet:dnSu(H) or *aox5*:Tet:dnCdc42 transgenes (Figure 5—figure supplement 5).”

Original: “stripes in zebrafish require a novel class of thin, fast cellular projection to transduce a Delta-Notch signal over long distances”

Revised: “stripes in zebrafish require a novel class of thin, fast cellular projection to promote transduction of a Delta-Notch signal over long distances”

Original: “We found that xanthoblasts sparsely populating prospective stripe regions, but not xanthophores of the interstripe, extend Delta+ airinemes that contact melanophores to activate Notch signaling, upregulate kita expression and promote consolidation into stripes (Figure 8-left).”

Revised: “Our data suggest a model in which xanthoblasts sparsely populating prospective stripe regions, but not xanthophores of the interstripe, extend Delta+ airinemes that contact melanophores, promoting Notch signaling and melanophore consolidation into stripes (Figure 6-left).”